# The AAA+ chaperone VCP disaggregates Tau fibrils and generates aggregate seeds in a cellular system

Itika Saha[1,2], Patricia Yuste-Checa [1,2], Miguel Da Silva Padilha[3,4,5], Qiang Guo [6,13], Roman Körner[1], Hauke Holthusen [1], Victoria A. Trinkaus[1,6,7], Irina Dudanova [3,4,5], Rubén Fernández-Busnadiego [2,6,8,9], Wolfgang Baumeister [6], David W. Sanders[10,14], Saurabh Gautam [1,15,16], Marc I. Diamond [10], F. Ulrich Hartl [1,2,7] ✉ & Mark S. Hipp [1,7,11,12] ✉

Amyloid-like aggregates of the microtubule-associated protein Tau are associated with several neurodegenerative disorders including Alzheimer's disease. The existence of cellular machinery for the removal of such aggregates has remained unclear, as specialized disaggregase chaperones are thought to be absent in mammalian cells. Here we show in cell culture and in neurons that the hexameric ATPase valosin-containing protein (VCP) is recruited to ubiquitylated Tau fibrils, resulting in their efficient disaggregation. Aggregate clearance depends on the functional cooperation of VCP with heat shock 70 kDa protein (Hsp70) and the ubiquitin-proteasome machinery. While inhibition of VCP activity stabilizes large Tau aggregates, disaggregation by VCP generates seeding-active Tau species as byproduct. These findings identify VCP as a core component of the machinery for the removal of neurodegenerative disease aggregates and suggest that its activity can be associated with enhanced aggregate spreading in tauopathies.

Deposition of amyloid-like Tau aggregates is a hallmark of devastating neurodegenerative disorders such as Alzheimer's disease and fronto-temporal dementia[1]. In healthy neurons, Tau functions in microtubule (MT) assembly and stabilization by associating with MTs via its repeat domain (RD) consisting of three or four 31-32 residue imperfect repeats. Two hexapeptide motifs within the RD are critical for Tau aggregation, and the RD forms the structural core of disease-associated aggregates[1], with RD mutations underlying familial tauopathies[2]. Expression of human Tau mutants in mouse models recapitulates essential features of tauopathies including the formation of amyloid-like Tau deposits and neuronal loss[3–5], indicating that Tau aggregation is central to neurodegeneration. Pathogenic Tau aggregates often exhibit the ability to induce aggregation in naïve cells through a mechanism of transcellular propagation that allows aggregate pathology to spread across brain regions[6,7]. Notably, pathological Tau aggregates and spreading are resolved upon lowering Tau levels,

which is accompanied by improved neuronal health and extended lifespan[8,9]. However, the cellular mechanisms involved in the reversal, clearance, and spread of Tau aggregates remain poorly understood.

While specialized chaperones of the AAA + family in bacteria, yeast, and plants have the ability to resolve amyloid-like aggregates[10,11], direct homologues of these hexameric disaggregases have not been identified in mammalian cells. Instead, disaggregation in higher eukaryotes is mainly attributed to the Hsp70 chaperone machinery[12–15]. The human Hsp70-Hsp40-Hsp110 chaperone system efficiently dissociates Tau and α-synuclein fibrils in vitro[16–18] independent of AAA + disaggregases that cooperate with the Hsp70 system in yeast and bacteria to achieve disaggregation[10]. The eukaryotic AAA + ATPase valosin-containing protein (VCP) exerts ATP-dependent protein unfolding activity[19,20] and has been proposed to resolve protein aggregates[21,22] and certain condensates such as stress granules[23,24]. VCP facilitates protein turnover via the ubiquitin-proteasome system[25,26], in

addition to sustaining functional autophagy[27]. Indeed, VCP mutations have been associated with aggregate deposition disorders such as vacuolar tauopathy and inclusion body myopathy associated with Paget disease of bone and frontotemporal dementia (IBMPFD)[21,28–30]. Accumulation of Tau aggregates in vacuolar tauopathy was proposed to be a consequence of diminished ATPase activity of mutant VCP (D395G)[21]. IBMPFD-associated VCP mutants exhibit increased basal ATP hydrolysis and unfolding activity[31,32], altered interactions with cofactors[33,34] and perturbed autophagic function[35]. However, VCP levels are not reduced in the brains of AD patients without these mutations[36], and the answer to the question of whether any of these mutations influence the clearance of pre-formed fibrillar Tau aggregates in cells is not known.

Here we provide direct evidence in a cell culture model and in primary murine neurons that VCP disaggregates amyloid-like Tau fibrils in a ubiquitin and proteasome-dependent manner, with the Hsp70 chaperone system contributing to aggregate clearance. This function of VCP is not detectably perturbed by pathogenic VCP mutations. Although disaggregation by VCP is coupled to proteasomal degradation, intermediates of the disaggregation process escape proteolysis, providing a possible source of seeding-competent Tau species.

## Results

To investigate the ability of cells to clear Tau aggregates, we used HEK293 cells stably expressing TauRD-Y (P301L/V337M), a mutant of the amyloid-forming repeat domain of Tau[37,38] fused to YFP via a flexible linker[39] (Fig. 1a). TauRD-Y is soluble and diffusely distributed in TauRD-Y cells, but the extracellular addition of Tau aggregates isolated from tauopathy brain tissue or generated in vitro induces its aggregation via template-based seeding, leading to the formation of aggregates that are stably propagated for weeks[39,40] (Fig. 1b). Using TauRD-Y aggregate seeds[39], we generated a cell line (TauRD-Y*) in which phosphorylated TauRD-Y accumulated in cytosolic inclusions 0.5–5 μm$^2$ in size (Supplementary Fig. 1a–c). TauRD-Y aggregates were also able to induce aggregates of full-length Tau fused to YFP (FLTau-Y). These aggregates reacted with the AT-8 antibody specific for phosphorylation at serine 202 and threonine 205 (epitopes not present in TauRD) (Fig. 1a, Supplementary Fig. 1d, e), which has been used previously to detect paired helical filaments[41,42]. Inclusions in TauRD-Y* cells stained with the amyloid-specific dye Amylo-Glo[43] (Fig. 1c). Analysis of the inclusions in intact TauRD-Y* cells by cryo-electron tomography (Supplementary Fig. 2a) revealed TauRD-Y fibrils of ~18 nm diameter, which were distinguishable from cytoskeletal structures (Fig. 1d) and consistent with the structures of fibrillar Tau in tauopathy patient brain[44–48]. Thus, TauRD forms amyloid-like fibrillar aggregates in TauRD-Y* cells.

### Proteasomal clearance of Tau aggregates

Soluble TauRD-Y was efficiently degraded in TauRD-Y cells upon inhibition of protein synthesis with cycloheximide (CHX) (Supplementary Fig. 3a). CHX treatment also led to partial clearance of TauRD inclusions and aggregated TauRD-Y in TauRD-Y* cells (Fig. 1e, Supplementary Fig. 3b). To avoid global inhibition of protein synthesis, we employed cells in which the expression of TauRD-Y is controlled with a Tet-regulated promoter (Tet-TauRD-Y and Tet-TauRD-Y* cells)[39]. The addition of doxycycline resulted in the clearance of TauRD-Y inclusions and insoluble TauRD-Y protein ($t_{1/2}$ ~12 h) (Supplementary Fig. 3c–f). The amount of insoluble TauRD-Y decreased faster than the level of soluble TauRD-Y (Supplementary Fig. 3f), consistent with aggregate material being solubilized prior to degradation. Moreover, inhibition of TauRD-Y synthesis resulted in a time-dependent reduction of inclusion size and number per cell (Fig. 1f). Thus, the cells are able to efficiently dissociate and degrade amyloid-like TauRD-Y aggregates.

The addition of the selective proteasome inhibitor Epoxomicin or siRNA-mediated downregulation of the proteasome component PSMD11 stabilized aggregated TauRD-Y upon doxycycline shut-off and prevented aggregate clearance (Supplementary Fig. 4a–e). Proteasome inhibition also stabilized soluble TauRD-Y in Tet-TauRD-Y cells[49] (Supplementary Fig. 4a), but did not lead to de novo Tau aggregation[50] (Supplementary Fig. 5i). Hence, the persistence of TauRD-Y aggregates upon proteasome inhibition is due to stabilization of pre-existing aggregates. In contrast, inhibition of lysosomal degradation, including micro-, macro- and chaperone-mediated autophagy, with Bafilomycin A1 (confirmed by increased levels of LC3-II) or inhibition of macro-autophagy with 3-methyladenine was without effect on the levels of total or aggregated TauRD-Y protein in the cellular model used (Supplementary Fig. 4a–c). Downregulation of autophagy components ATG5/7 supported this conclusion (Supplementary Fig. 4d, e). Thus, TauRD-Y aggregates are degraded in a proteasome-dependent, autophagy-independent manner in our model system.

### Tau disaggregation requires VCP

Proteins must generally be unfolded to access the catalytic center of the 20 S proteasome. Thus, prior to degradation, aggregated proteins need to undergo disaggregation[51]. To identify the cellular machinery involved in TauRD-Y disaggregation, we performed an interactome analysis of aggregated TauRD-Y by quantitative mass spectrometry. We identified the AAA + ATPase VCP as one of the most highly enriched interactors of aggregated TauRD-Y, along with the ubiquitin-binding VCP cofactors UFD1L, NPLOC4, and NSFL1C, and multiple subunits of the 26 S proteasome (Fig. 2a, Supplementary Table 1). Hsp70 was detected in the proteomic analysis but was not enriched on aggregated TauRD-Y. Co-localization of VCP and its cofactors with TauRD-Y aggregates was confirmed by fluorescence microscopy (Fig. 2b, Supplementary Fig. 5a–c).

VCP utilizes the energy from ATP hydrolysis to structurally remodel and unfold proteins in different cellular contexts[19,52]. To assess whether VCP is involved in TauRD-Y disaggregation, we inhibited VCP in cells using NMS-873, a small molecule allosteric inhibitor of the VCP ATPase[53]. Similar to proteasome inhibition, NMS-873 blocked the clearance of TauRD-Y aggregates when TauRD-Y synthesis was stopped with doxycycline (Fig. 2c, d). Likewise, the aggregates were stabilized when VCP was inhibited using CB-5083 (Supplementary Fig. 5d, e), a competitive inhibitor of ATP binding in the D2 ATPase domain of VCP[54], or down-regulated with siRNA (Supplementary Fig. 5f, g). VCP inhibition during ongoing TauRD-Y synthesis resulted in a significant increase in inclusion size (Supplementary Fig. 5h), suggesting that the inclusions exist at a dynamic equilibrium between formation and disaggregation. No aggregation of soluble TauRD-Y was detected after treating cells with NMS-873 or VCP siRNA (Supplementary Fig. 5i, j). VCP down-regulation caused a marginal increase in the level of soluble TauRD-Y in Tet-TauRD-Y cells, but did not result in a significant stabilization after doxycycline addition (Supplementary Fig. 5k). In contrast, aggregate-containing Tet-TauRD-Y* cells treated with VCP siRNA accumulated significantly higher amounts of TauRD-Y both in the absence or presence of doxycycline, indicating an aggregate-specific role of VCP (Supplementary Fig. 5k).

Importantly, VCP also co-localized with aggregates of full-length Tau (FLTau-Y) in FLTau-Y* cells (Supplementary Fig. 6a), and VCP or proteasome inhibition prevented the clearance of FLTau-Y aggregates (Supplementary Fig. 6b, c), recapitulating the behavior of TauRD. To exclude a possible role of the YFP tag on Tau in VCP-mediated disaggregation, we generated HEK293T cells stably expressing non-tagged full-length Tau (FLTau) and myc-tagged Tau repeat domain (TauRD) under a Tet-regulated promoter (Tet-FLTau, Tet-FLTau* and Tet-TauRD, Tet-TauRD* cells). Similar to FLTau-Y aggregates in FLTau-Y* cells, FLTau aggregates in Tet-FLTau* cells were phosphorylated at serine 202 and threonine 205 and colocalized with VCP (Fig. 2e,

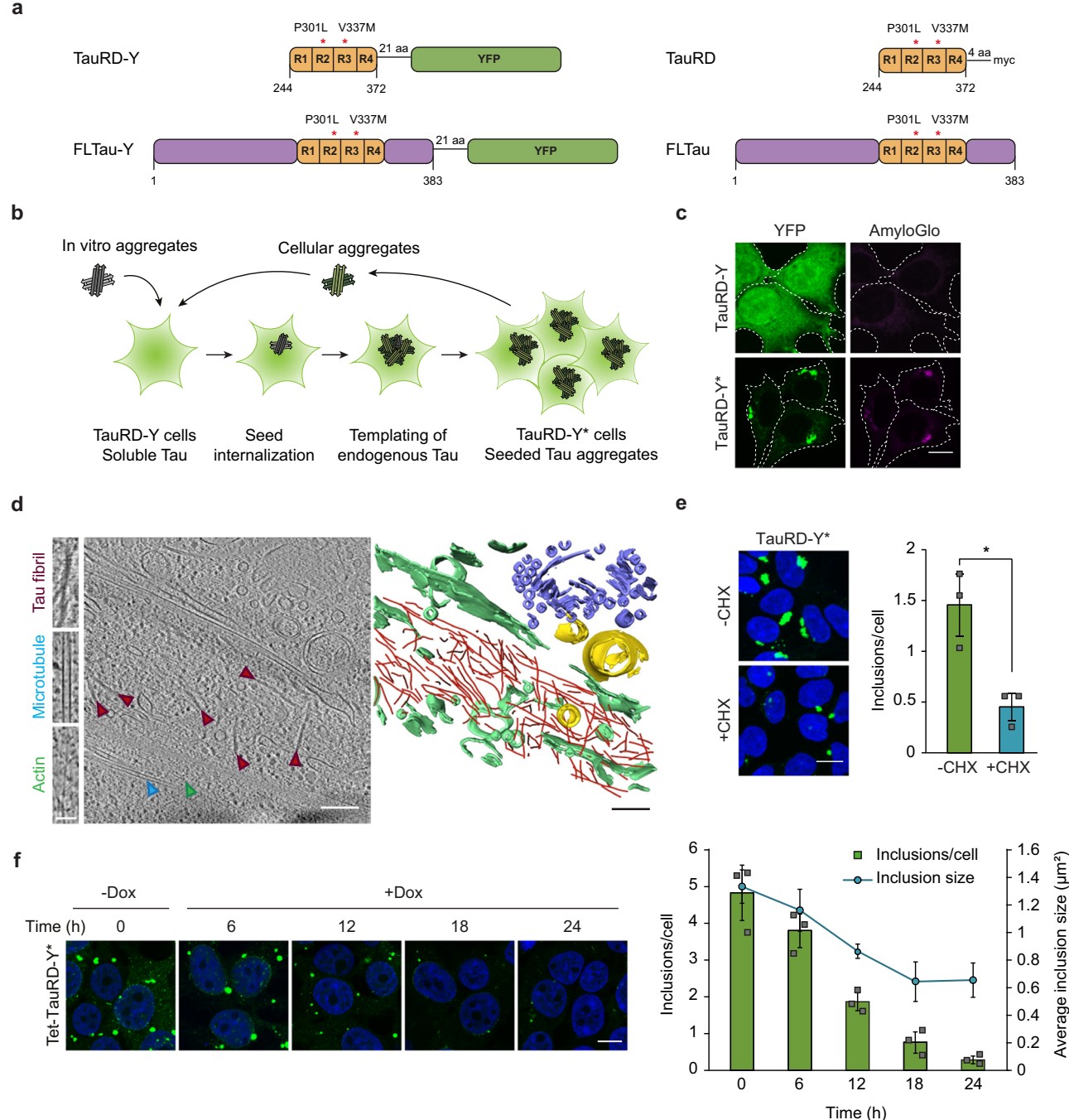

**Fig. 1 | TauRD-Y forms amyloid-like aggregates that are cleared from cells.**
**a** Schematic representation of Tau constructs used in this study. TauRD-Y, the repeat domain, and FLTau-Y, the 0N4R isoform of full-length (FL) Tau with two frontotemporal dementia-associated mutations, P301L and V337M, fused to YFP via 21 amino acid (aa) linkers or without YFP. **b** Schematic representation of aggregate seeding. Extracellular addition of preformed Tau aggregates induces templating of intracellular Tau into aggregates that propagate with cell division. Aggregate seeds may be generated in vitro or contained in cell lysate. TauRD-Y, naïve cells containing soluble TauRD-Y; TauRD-Y*, cells containing TauRD-Y aggregates. **c** Staining of TauRD-Y and TauRD-Y* cells with the amyloid-specific dye Amylo-Glo (magenta). White dashed lines indicate cell boundaries. Scale bar, 10 μm. **d** TauRD-Y aggregates are fibrillar in structure. Left, a 1.7 nm thick tomographic slice of a TauRD inclusion

from TauRD-Y* cells is shown. Red, blue and green arrowheads indicate representative TauRD-Y fibril, microtubule and actin, respectively. Right, 3D rendering of corresponding tomogram showing TauRD-Y fibrils (red), Golgi (purple), mitochondria (yellow) and ER (green). Scale bars, 200 nm, inset 40 nm. **e** Aggregate clearance. Left, TauRD-Y* cells were treated for 24 h with cycloheximide (CHX; 50 μg/mL) where indicated. Nuclei were counterstained with DAPI (blue). Scale bar, 10 μm. Right, quantification of TauRD-Y foci. Mean ± s.d.; $n = 3$; 500–600 cells analyzed per experiment; *$p < 0.05$ ($p = 0.0151$) from two-tailed Student's paired t-test. **f** Left, representative images of Tet-TauRD-Y* cells treated with doxycycline (Dox; 50ng/mL) for the indicated times. Right, quantification of inclusions per cell and average inclusion size (μm²). Mean ± s.d.; $n = 3$. Scale bar, 10 μm. Source data are provided as a Source Data file.

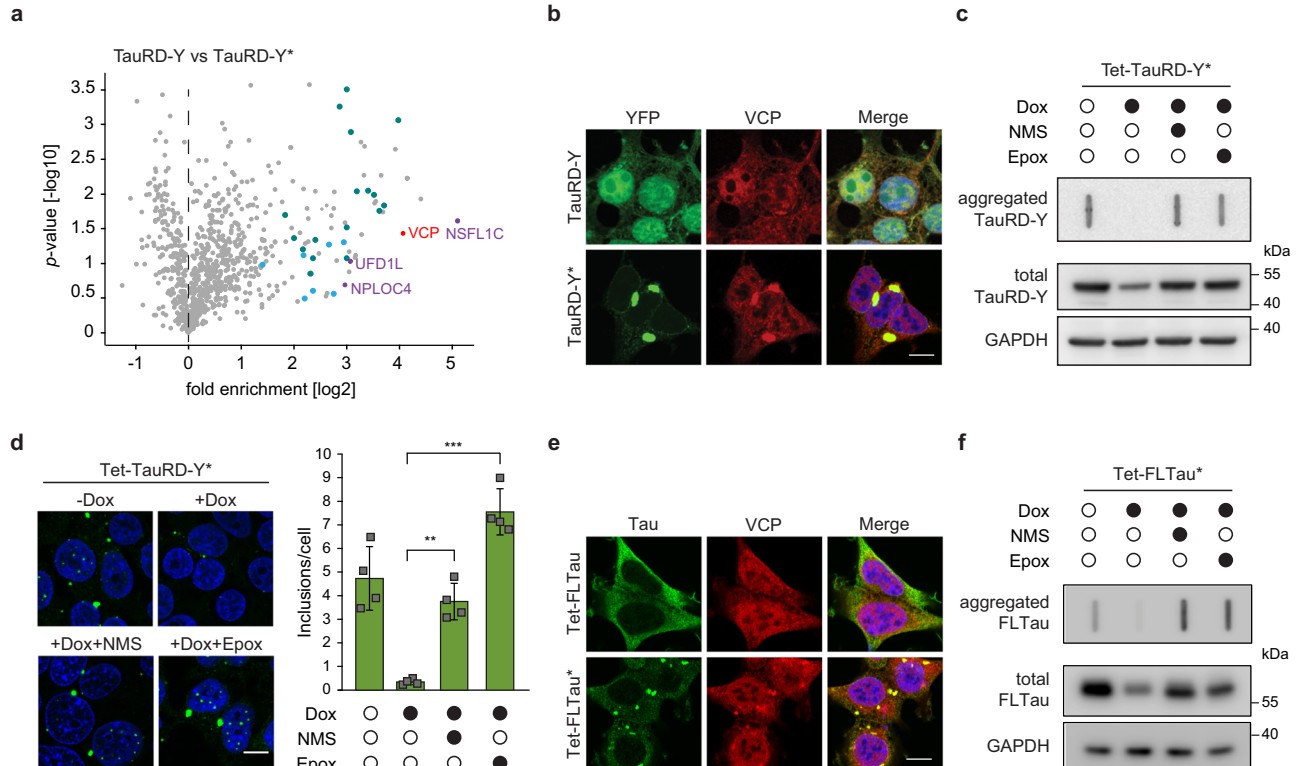

**Fig. 2 | Disaggregation of Tau aggregates is dependent on VCP activity.**
**a** Volcano plot of TauRD-Y interactome from TauRD-Y and TauRD-Y* cells. Unlabeled green and blue symbols represent proteasome subunits of 19 S and 20 S, respectively. VCP and its cofactors are highlighted. P-values from one-sample test. **b** Association of VCP with TauRD-Y inclusions. Immunofluorescence staining of VCP (red) and YFP fluorescence of TauRD-Y (green) in TauRD-Y and TauRD-Y* cells. Representative images from at least three independent experiments are shown. Scale bar, 10 μm. **c** Filter trap analysis of lysates from Tet-TauRD-Y* cells treated for 24 h with doxycycline (Dox; 50 ng/mL) alone or in combination with NMS-873 (NMS; 2.5 μM) or Epoxomicin (Epox; 50 nM). Aggregated and total TauRD-Y levels were determined by immunoblotting against GFP. GAPDH served as loading control. Representative blots from at least three independent experiments are shown. **d** Left, representative images of Tet-TauRD-Y* cells treated as in (**c**). Scale bar,

10 μm. Right, quantification of the number of TauRD-Y inclusions per cell. Mean ± s.d.; $n = 4$; 300–600 cells analyzed per experiment. $*p < 0.05$ (+ Dox vs + Dox + NMS, $p = 0.0019$); $**p < 0.01$ (+ Dox vs + Dox + Epox, $p = 0.0004$) from two-tailed Student's paired $t$-test. **e** Association of VCP with FLTau inclusions. Immunofluorescence staining of VCP (red) and Tau with Tau-5 antibody (green) in FLTau and FLTau* cells. Representative images from three independent experiments are shown. Scale bar, 10 μm. **f** Filter trap analysis of lysates from Tet-FLTau* cells treated for 24 h with doxycycline (Dox; 50 ng/mL) alone or in combination with NMS-873 (NMS; 2.5 μM) or Epoxomicin (Epox; 50 nM). Aggregated and total FLTau levels were determined by immunoblotting using AT8 and Tau-5 antibodies, respectively. GAPDH served as the loading control. Representative blots from at least three independent experiments are shown. Source data are provided as a Source Data file.

Supplementary Fig. 6d). FLTau and TauRD aggregates were resolved in a VCP and proteasome-dependent manner when Tau synthesis was halted by adding doxycycline (Fig. 2f, Supplementary Fig. 6e).

We next tested whether VCP also modulates Tau aggregation in neurons. Mouse primary neurons were transduced to express soluble TauRD-Y (Fig. 3a, b). Upon seeding with TauRD aggregates[39], we observed the formation of multiple inclusions of intracellular TauRD-Y (Fig. 3b). Cryo-electron tomography of TauRD-Y inclusions in aggregate-containing neurons revealed fibrillar aggregates similar to the aggregates in TauRD-Y* cells (Fig. 3c, Supplementary Fig. 2). The lower density of the neuronal cytosol allowed the observation that the TauRD-Y fibrils were coated with globular domains consistent with the presence of YFP on TauRD (Supplementary Fig. 6f), as previously observed for other amyloidogenic proteins fused to a fluorescent protein[55,56]. Aggregate seeding in neurons was accompanied by a ~40% decrease in cell viability (Fig. 3d). Most of the neuronal TauRD-Y inclusions stained positive for VCP (Fig. 3b). Treatment with the VCP inhibitor NMS-873 for 4 h caused a massive accumulation of TauRD aggregates in seeded neurons, in some cases occupying most of the cell body area (Fig. 3e). No inclusions were observed in unseeded cells upon VCP inhibition (Fig. 3e). These results demonstrate that VCP functions in TauRD-Y disaggregation in neurons. To test whether VCP interacts with Tau aggregates in an animal model for tauopathy, we

used the mouse model rTg4510 that expresses mutant human Tau P301L in the forebrain[5]. At 4 months of age, these mice develop tangle-like inclusions that progress in an age-dependent manner[5]. Immunofluorescence analysis of brain sections from 4- and 16-month-old mice revealed that VCP colocalized with Tau inclusions in 49.5 ± 14.1% of the neurons that contained phosphorylated Tau, whereas VCP remained diffusely distributed in control mice (Supplementary Fig. 6g). This is consistent with previous reports that VCP is strongly enriched in the SDS-insoluble fractions of rTg4510 mice forebrain[57]. Taken together these findings suggest that VCP has a critical role in the disposal of aggregated Tau in vivo.

**Disaggregation depends on substrate ubiquitylation**
Ubiquitylation of VCP substrates, particularly the formation of lysine 48 (K48) linked polyubiquitin chains, is required for VCP recruitment[19,20,31,52]. We, therefore, analyzed immunoprecipitates of TauRD-Y for the presence of ubiquitin. Only in TauRD-Y* cells containing aggregated TauRD-Y was the protein detectably modified by the addition of 1 to 4 ubiquitin molecules (Fig. 4a). Analysis with a K48-specific antibody verified the presence of K48-linked ubiquitin (Fig. 4a). Immunofluorescence imaging also showed that the TauRD-Y aggregates stained positive for K48-linked ubiquitin chains (Fig. 4b, Supplementary Fig. 7a), while K63-linked ubiquitin was not detectable

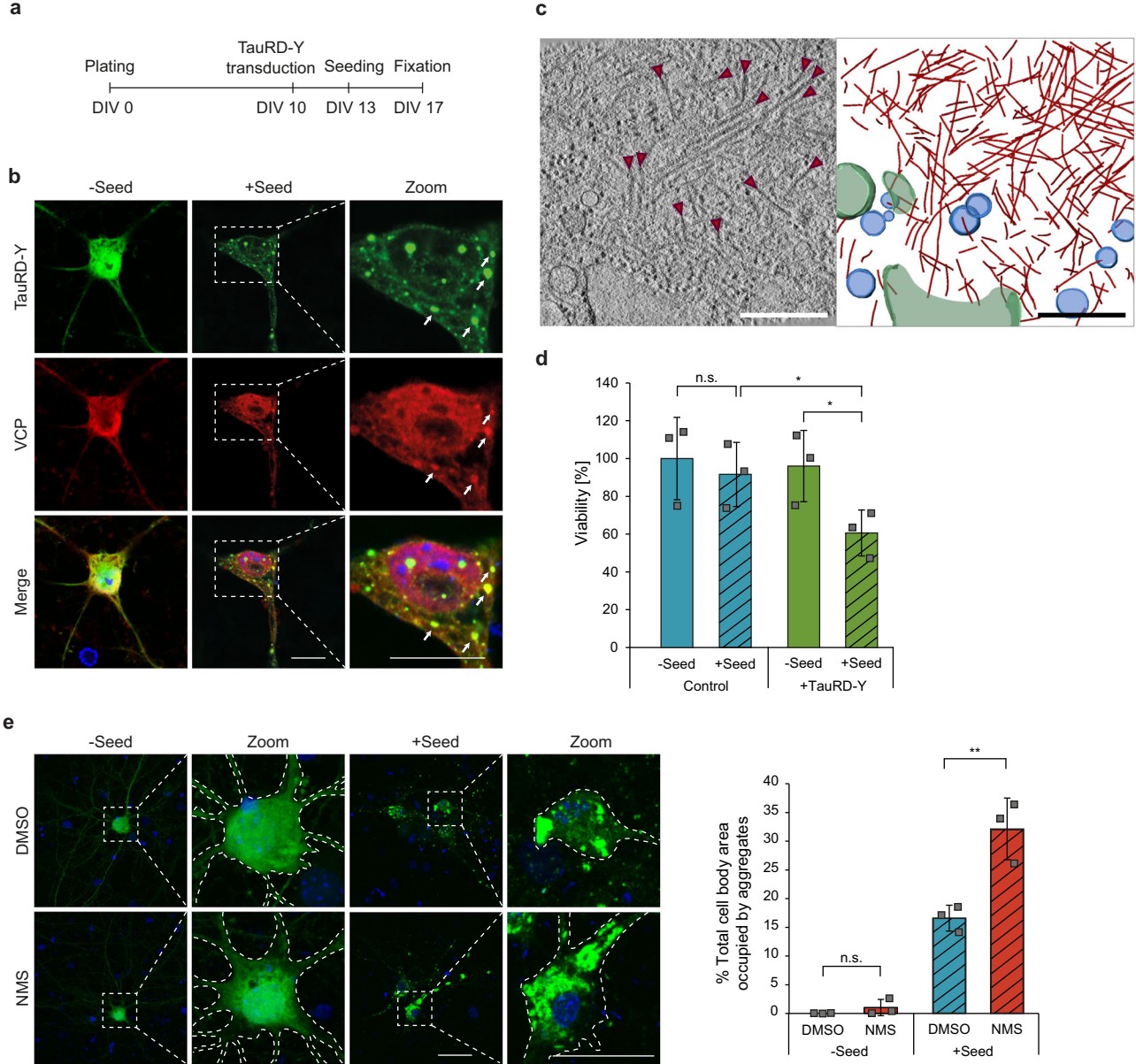

**Fig. 3 | Disaggregation of TauRD-Y aggregates in primary neurons is dependent on VCP activity. a** Schematic representation of experimental timeline in primary neurons. DIV, days in vitro. **b** Association of VCP with TauRD-Y inclusions in primary neurons. Immunofluorescence staining of VCP (red) and YFP fluorescence of TauRD-Y (green). Arrows point to TauRD-Y inclusions containing VCP. Representative images from three independent experiments are shown. Scale bars, 10 μm. **c** Fibrillar TauRD-Y aggregates in primary neurons. Left, a 1.4 nm thick tomographic slice of a TauRD inclusion from neurons is shown. Red arrowheads indicate TauRD-Y fibrils. Right, 3D rendering of corresponding tomogram showing TauRD-Y fibrils (red), vesicles (blue), and ER (green). A representative tomogram from two independent experiments is shown. Scale bar, 350 nm. **d** Toxicity of TauRD-Y aggregation in primary neurons. Untransduced neurons or neurons

transduced with TauRD-Y were treated with cell lysates containing TauRD-Y aggregates where indicated. Viability was measured 4 days later using an MTT assay. Mean ± s.d.; $n = 3$; $*p < 0.05$ (Control + Seed vs TauRD-Y + Seed, $p = 0.0184$; TauRD-Y - Seed vs TauRD-Y + Seed, $p = 0.0142$); n.s. non-significant (Control - Seed vs Control + Seed, $p = 0.2074$) from two-way ANOVA with Tukey post hoc test. **e** Left, representative images of primary neurons expressing TauRD-Y, exposed to cell lysates containing TauRD-Y aggregates and treated for 4 h with NMS-873 (NMS; 0.5 μM) where indicated. Dashed lines indicate the contours of the cells. Scale bars, 20 μm. Right, quantification of area occupied by TauRD-Y aggregates as a percentage of the cell body area. Mean ± s.d.; $n = 3$; $**p < 0.01$ (+ Seed + DMSO vs + Seed + NMS, $p = 0.0098$); n.s. non-significant (- Seed + DMSO vs - Seed + NMS, $p = 0.2998$) from unpaired t test. Source data are provided as a Source Data file.

(Supplementary Fig. 7a). Likewise, the TauRD-Y inclusions in primary neurons colocalized with poly-ubiquitin chains (Supplementary Fig. 7b). K48-linked ubiquitin signal was also observed on the aggregates of untagged FLTau and myc-tagged TauRD (Supplementary Fig. 7c).

Inhibition of the ubiquitin-activating enzyme E1 with the specific inhibitor MLN7243[58] efficiently blocked ubiquitin conjugation (Supplementary Fig. 8a). TauRD-Y inclusions were still present but were no longer ubiquitin K48-reactive (Fig. 4b, Supplementary Fig. 8b).

VCP was not recruited to these aggregates (Fig. 4c, Supplementary Fig. 8c), and both disaggregation and degradation of TauRD-Y in Tet-TauRD-Y* cells were blocked (Fig. 4d, Supplementary Fig. 8d). MLN7243 treatment also prevented the degradation of soluble TauRD-Y to a degree similar to proteasome inhibition (Supplementary Fig. 8e). Together these data show that VCP recruitment requires ubiquitylation of aggregated Tau, followed by disaggregation and remodeling to species that are accessible for proteasomal degradation.

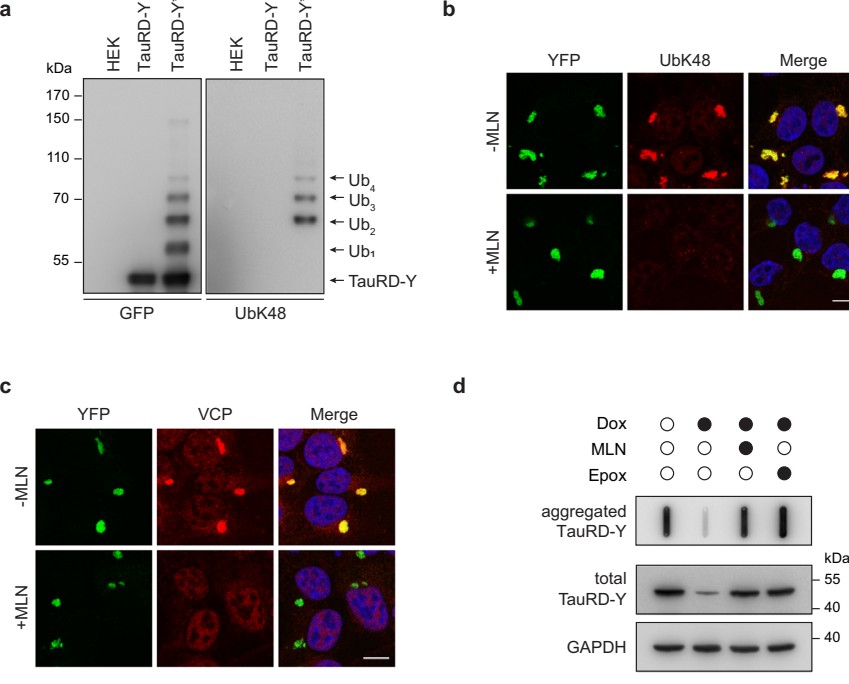

**Fig. 4 | Ubiquitination is necessary for VCP recruitment and disaggregation.** **a** Immunoprecipitation of TauRD-Y from lysates of control HEK cells, TauRD-Y, and TauRD-Y* cells in the presence of 0.1% SDS using anti-GFP beads. Eluates were analyzed by immunoblotting with antibodies against GFP and K48-linked ubiquitin chains (UbK48). The TauRD-Y band shows the unmodified protein and arrowheads point at increments in ubiquitin conjugation ($Ub_1$-$Ub_4$). Representative blots from two independent experiments are shown. **b** Inhibition of ubiquitylation of TauRD inclusions. TauRD-Y* cells were treated for 12 h with MLN7243 (MLN; 0.5 μM) followed by immunofluorescence analysis with a UbK48 antibody (red).

Representative images from two independent experiments are shown. **c** Inhibition of TauRD ubiquitylation prevents VCP association. TauRD-Y* cells were treated as in (**b**). VCP (red) was visualized by immunofluorescence. Representative images from two independent experiments are shown. Scale bars, 10 μm. **d** Filter trap analysis of lysates from Tet-TauRD-Y* cells treated for 24 h with 50 ng/mL doxycycline alone or in combination with 0.2 μM MLN7243 or 50 nM Epoxomicin. Aggregated and total TauRD-Y levels were determined by immunoblotting against GFP. GAPDH served as a loading control. Representative blots from two independent experiments are shown.

## Functions of VCP and Hsp70 in disaggregation

Disaggregation of both heat stress-induced and amyloid-like aggregates in mammalian cells has been assigned to the Hsp70 chaperone system[12–14,17]. Our findings raised the possibility of functional cooperation between VCP and Hsp70 in these processes. To determine whether VCP participates in dissolving heat-induced aggregates, we expressed the metastable protein firefly luciferase (Fluc) fused to GFP in HEK293 cells. Heat stress at 43 °C for 2 h combined with proteasome inhibition resulted in the formation of large (~2–3 μm) Fluc-GFP inclusions[59] (Supplementary Fig. 9a). Unlike the TauRD-Y inclusions, the Fluc-GFP aggregates did not stain with an amyloid-specific dye (Supplementary Fig. 9a), suggesting that they were amorphous in structure. The Fluc-GFP aggregates were ubiquitin-negative and did not co-localize with VCP (Supplementary Fig. 9b, c). Accordingly, VCP inhibition with NMS-873 did not interfere with disaggregation (Supplementary Fig. 9d), arguing against a role of VCP in this process. However, inhibition of the ATPase activity of Hsp70 with the inhibitor VER-155008[60] prevented Fluc-GFP disaggregation (Supplementary Fig. 9d), confirming the role of the Hsp70 system in disaggregation.

To investigate whether Hsp70 participates in TauRD-Y disaggregation, we treated Tet-TauRD-Y* cells with VER-155008 or with NMS-873 and stopped TauRD-Y synthesis with doxycycline. VCP inhibition stabilized both large (>1.5 μm²) and small (<1.5 μm²) TauRD-Y inclusions (Supplementary Fig. 10a, b). In contrast, Hsp70 inhibition stabilized large aggregates only partially and resulted in a marked accumulation of small inclusions, consistent with VCP acting before Hsp70 in the disaggregation process (Supplementary Fig. 10a, b). These findings suggested that Hsp70 cooperates with VCP in disaggregation, either by dissociating fragments generated by VCP and/

or by preventing re-aggregation of TauRD liberated from inclusions by VCP. Since Hsp70 was not enriched on TauRD-Y aggregates in the proteomic analysis (Supplementary Table 1), its interaction with TauRD may be transient.

## Effects of VCP mutants on Tau disaggregation

Point mutations in VCP are associated with dominantly inherited disorders such as Inclusion body myopathy with Paget disease of bone and frontotemporal dementia (IBMPFD)[27] and vacuolar tauopathy[21]. These mutations lead to a dominant negative loss or alteration of VCP function, presumably due to the oligomeric nature of VCP[61,62]. The mutation D395G (DG), associated with vacuolar tauopathy is located in the D1 ATPase domain of VCP (Fig. 5a). It has been reported to have a mildly reduced capacity to disaggregate Tau fibrils in an in vitro system, due to a ~30% reduced ATPase activity[21]. The IBMPFD mutations, A232E (AE) and R155H (RH), are located in the D1 ATPase domain and in the N-domain, respectively, and are associated with enhanced ATPase activity compared to wild type (WT) VCP[32] (Fig. 5a). We tested whether these mutations impair Tau disaggregation in our cellular model using the ATPase defective VCP double-mutant E305Q/E578Q (EQ/EQ)[63] (Fig. 5a) as a control. The mutant proteins, carrying a C-terminal myc-tag, were transiently overexpressed in Tet-TauRD-Y* cells for 24 h, and then TauRD-Y synthesis was stopped with doxycycline to observe disaggregation. Note that mutant VCP was expressed in cells containing pre-formed aggregates to exclude a potential role of VCP in aggregate seeding[64]. The myc-tagged mutant proteins were present in hexamers that migrated on native PAGE like WT VCP (Supplementary Fig. 11a) and colocalized with TauRD-Y aggregates (Fig. 5b).

As expected, expression of the ATPase defective VCP (EQ/EQ) effectively prevented TauRD-Y aggregate clearance, even though the

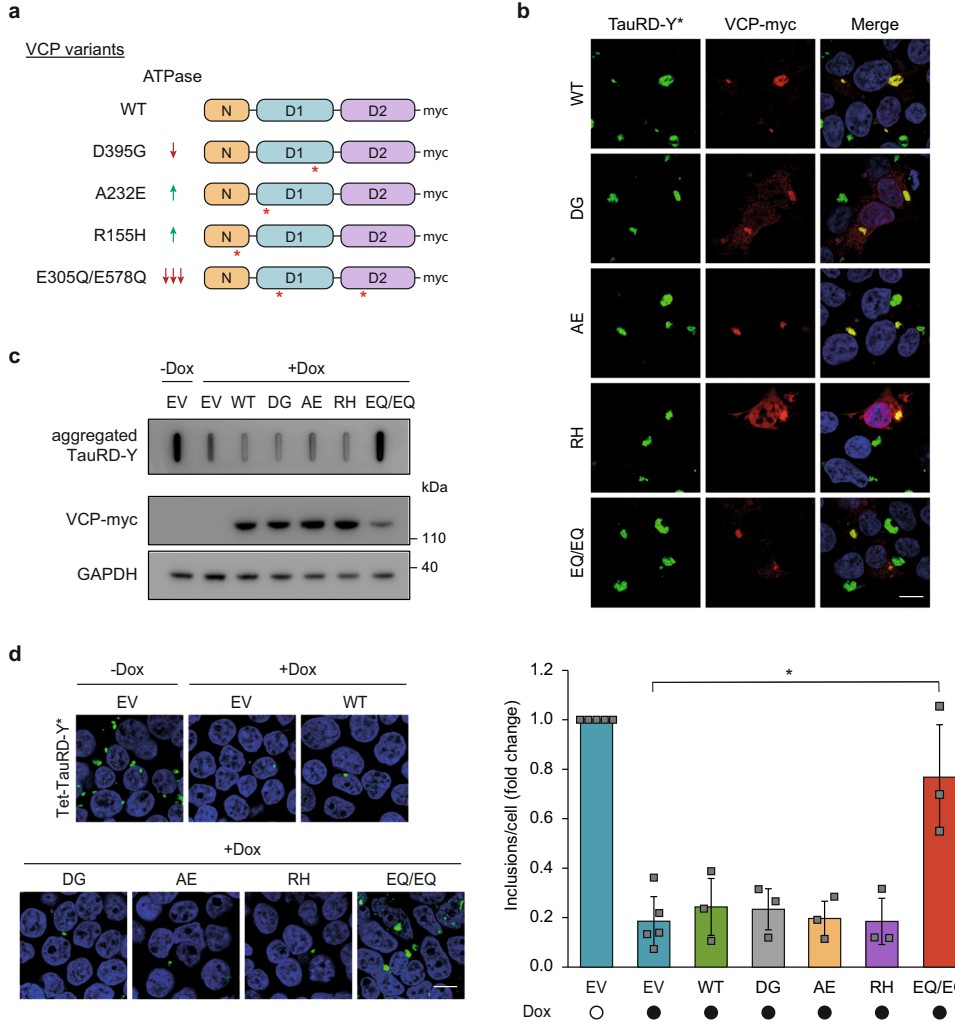

**Fig. 5 | Effects of VCP mutants on Tau disaggregation. a** Schematic representation of VCP variants used in this study. Wild type (WT), D395G (DG), A232E (AE), R155H (RH) and E305Q/E578Q (EQ/EQ) VCP were tagged with a C-terminal myc-tag. Red asterisks indicate relative positions of the mutations. **b** Association of transiently expressed VCP variants with TauRD-Y inclusions. Immunofluorescence staining of myc (red) and YFP fluorescence of TauRD-Y (green) in TauRD-Y* cells. Representative images from two independent experiments are shown. Scale bar, 10 μm. **c** Filter trap analysis of lysates from Tet-TauRD-Y* cells transiently transfected with empty vector (EV) or indicated VCP variants for 24 h, and treated for another 24 h with doxycycline (Dox; 50 ng/mL). Aggregated TauRD-Y and overexpressed VCP levels were determined by immunoblotting against GFP and myc, respectively. GAPDH served as loading control. Representative blots from three independent experiments are shown. **d** Left, representative images of Tet-TauRD-Y* cells treated as in (**c**). Scale bar, 10 μm. Right, quantification of aggregate foci. Mean ± s.d.; n = 3; > 400 cells analyzed per experiment; *p < 0.05 (EV + Dox vs EQ/EQ + Dox, p = 0.0209) from two-tailed Student's paired t-test. Source data are provided as a Source Data file.

expression levels of this mutant were relatively low when compared with the other constructs (Fig. 5c, d). However, none of the disease-related VCP mutants, including the vacuolar tauopathy mutant DG, when expressed at the indicated levels, detectably stabilized TauRD-Y aggregates (Fig. 5b–d). Similar results were obtained when the presence of aggregates was specifically analyzed in cells expressing the mutant VCP proteins by immunofluorescence (Supplementary Fig. 11b). In conclusion, the effect of the VCP disease mutations on disaggregation in HEK293 cells, if any, is only mild.

**VCP generates Tau species capable of seeding aggregation**
Progression of tauopathies and other neurodegenerative diseases is thought to be mediated by aggregate spreading across brain regions through a prion-like seeding mechanism[6,7]. We speculated that the disaggregation activity of VCP could modulate the levels of aggregate species that are able to induce the aggregation of soluble Tau in recipient cells. To address this possibility, we measured the presence of seeding-competent TauRD species by FRET in cells expressing TauRD-mTurquoise2 and TauRD-Y (TauRD-TY cells)[65] (Fig. 6a). Addition of aggregate-containing total lysates from control TauRD-Y* cells induced TauRD aggregation in reporter cells (Supplementary Fig. 12a, b). Strikingly, treatment of TauRD-Y* cells with the VCP inhibitor NMS-873 reduced the seeding capacity of lysates by more than 50%, when equivalent amounts of TauRD-Y were compared (Fig. 6b). In contrast, such a reduction in seeding was not observed when cells were treated with proteasome inhibitor (Epoxomicin) or Hsp70 inhibitor (VER-155008) (Fig. 6b). However, treatment with the E1 enzyme inhibitor MLN7243, which prevented VCP recruitment to the aggregates (Fig. 4c), also caused a ~50% reduction of FRET positive TauRD-TY reporter cells. Similar effects were observed when lysates from TauRD-Y* cells transiently expressing VCP EQ/EQ were used. In contrast, expression of VCP DG did not reduce seeding (Fig. 6c, Supplementary Fig. 12c). Thus, VCP-mediated disaggregation generates seeding-active TauRD-Y species.

To characterize the seeding competent material in the presence and absence of VCP function, we fractionated lysates from TauRD-Y*

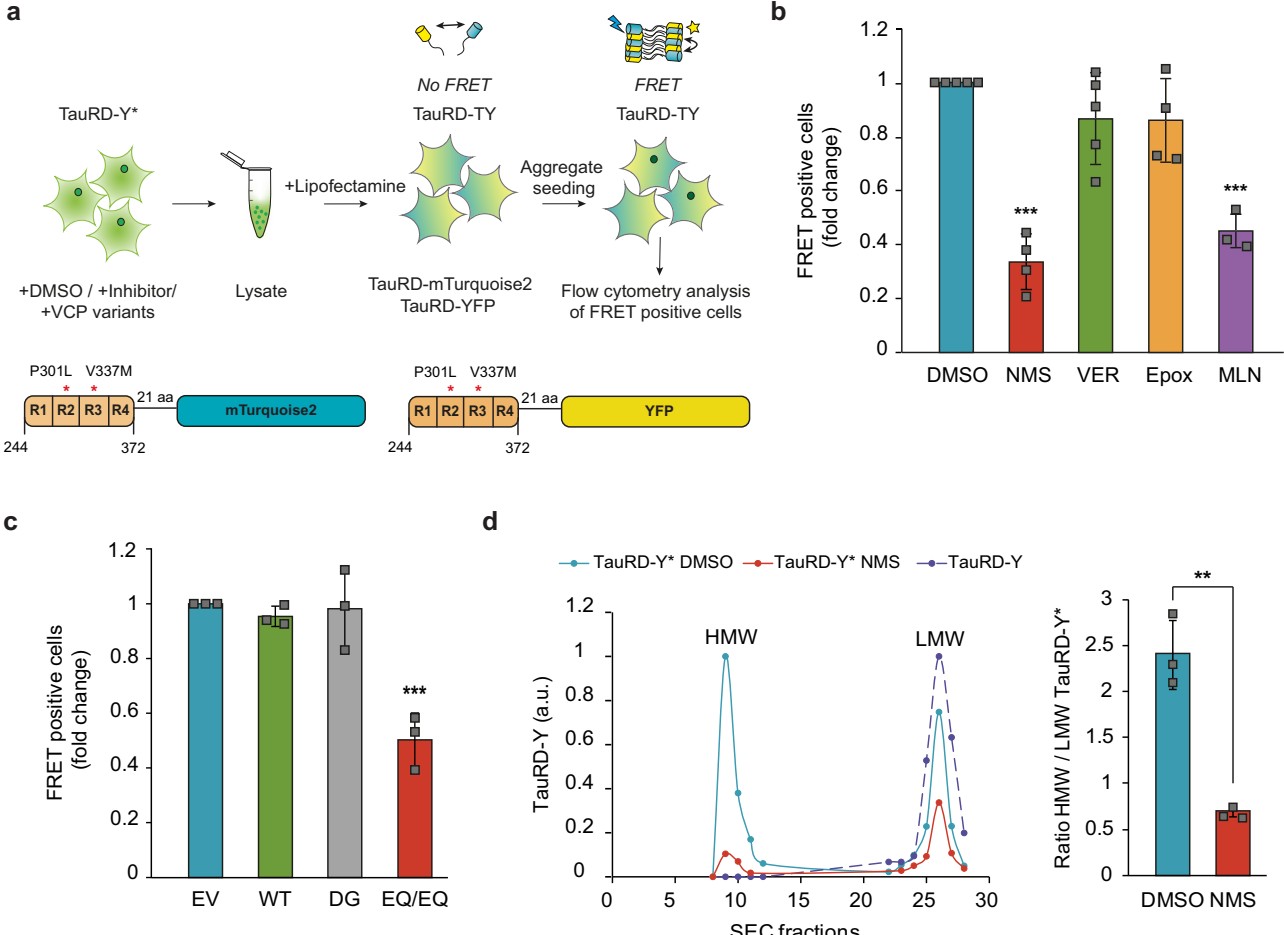

**Fig. 6 | VCP-mediated disaggregation generates seeding-competent TauRD-Y species. a** Experimental scheme to assess the effects of inhibitors of VCP, Hsp70, proteasome, and ubiquitylation on the level of TauRD-Y aggregate seeds in TauRD-Y* cells. **b** Flow cytometry analysis of aggregate seeding in TauRD-TY reporter cells after the addition of lysates from TauRD-Y* cells treated with NMS (VCP inhibitor), VER (Hsp70 inhibitor), Epox (proteasome inhibitor) and MLN (ubiquitylation inhibitor). Fold changes with respect to DMSO-treated cells are shown. Mean ± s.d.; NMS and Epox $n = 4$, VER $n = 5$, MLN $n = 3$; ***$p < 0.001$ (DMSO vs NMS, $p = 8.69 \times 10^{-7}$; DMSO vs MLN, $p = 4.2 \times 10^{-5}$) from one-way ANOVA with Tukey post hoc test. **c** Flow cytometry analysis of aggregate seeding in TauRD-TY reporter cells after addition of lysates from TauRD-Y* cells transfected with empty vector (EV), wild-type (WT), D395G (DG) and ATPase deficient E305Q/E578Q (EQ/EQ) VCP constructs. Fold changes with respect to EV transfected cells are shown. Mean ± s.d. $n = 3$; ***$p < 0.001$ (EV vs EQ/EQ, $p = 0.0007$) from one-way ANOVA with Tukey post hoc test. **d** Left, fractionation of TauRD-Y from DMSO and NMS-873 treated lysates of TauRD-Y* cells by size exclusion chromatography (SEC). Equal amounts of total lysate protein were analyzed. Y-axis represents the relative amount of TauRD-Y in the high molecular weight (HMW) and the low molecular weight (LMW) fractions quantified by immunoblotting. Right, the ratio of TauRD-Y in HMW/LMW fractions. Mean ± s.d.; $n = 3$. **$p < 0.01$ ($p = 0.002$) from two-tailed Student's paired $t$-test. Source data are provided as a Source Data file.

cells by size-exclusion chromatography. Inclusions >0.2 μm were removed by filtration. The majority of the remaining TauRD-Y (~70%) fractionated at a high molecular weight (HMW) of ≥40 MDa in the void volume of the column. The remainder fractionated at a low molecular weight (LMW), equivalent to the position of soluble TauRD-Y from TauRD-Y cells (Fig. 6d). Both fractions isolated from TauRD-Y* cells were seeding competent, but the specific seeding activity of HMW TauRD-Y (% FRET positive cells per ng TauRD-Y) was ~10-fold higher than that of the LMW fraction (Supplementary Fig. 12d). Treatment with VCP inhibitor NMS-873 strongly reduced the total amount of TauRD-Y species <0.2 μm, consistent with the reduced seeding activity after VCP inhibition. Moreover, the ratio between HMW and LMW peaks was reversed as the former was decreased by ~80% and the latter by only ~25% (Fig. 6d). However, the specific seeding activity of TauRD in both fractions remained unchanged (Supplementary Fig. 12d), suggesting that VCP inhibition reduces the amount of seeds but not their intrinsic seeding potency. Together these results demonstrate that the disaggregation activity of VCP increases the available pool of seeding competent TauRD species.

## Discussion

Metazoa do not possess a homologue of the AAA + ATPase Hsp104 responsible for protein disaggregation in bacteria, fungi and plants[11,66]. Instead, dissociation of large protein aggregates, including amyloid fibrils, in animal cells is generally ascribed to the Hsp70/Hsp110/Hsp40 chaperone system[11,13–15,17]. Here we provide evidence that the AAA + ATPase VCP functions in disaggregating amyloid fibrils of Tau in human cells and primary mouse neurons (Fig. 7). VCP is distinct from Hsp104 in that it requires the target aggregate to be ubiquitylated, a critical element of control to ensure specificity and avoid dissolution of functional protein assemblies[67]. Consistent with such a control function, ubiquitylation likely occurs after aggregate formation as an essential prerequisite for disaggregation (Fig. 7). The E3 ubiquitin ligases involved in this process remain to be identified. Moreover, aggregate ubiquitylation ensures that disaggregation by VCP is coupled to degradation by the 26 S proteasome. In addition, the proteasomal 19 S ATPase may contribute to disaggregation, consistent with its ability to fragment fibrils in vitro[68]. The Hsp70 chaperone system is required for the overall efficiency of the reaction, either by further

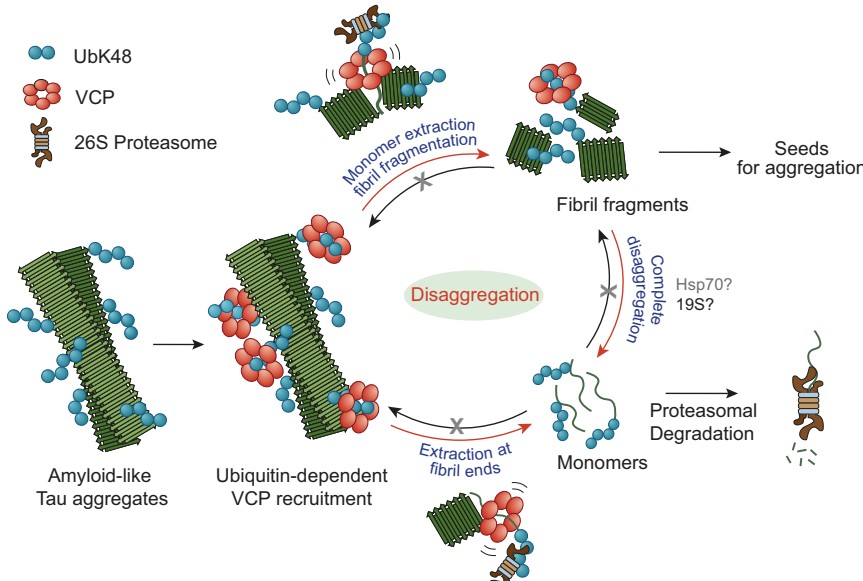

**Fig. 7 | Model of VCP-mediated disaggregation of amyloid-like Tau aggregates.** Modification of aggregates with K48-linked ubiquitin chains allows the recruitment of VCP. VCP may extract ubiquitylated Tau monomer from fibril ends or from within fibrils. Monomers are directly targeted for proteasomal degradation. Extraction from internal sites results in fibril fragmentation and the generation of oligomers that act as seeds for aggregation. Completion of oligomer disaggregation may be accomplished by the 19 S proteasome, perhaps with the participation of Hsp70 system (grey). Hsp70 may also contribute to aggregate clearance by preventing re-aggregation of disaggregation products.

dissociating aggregate fragments produced by VCP action or by preventing re-aggregation of Tau molecules that have been liberated from the fibrils (Fig. 7). As the smaller TauRD aggregates that accumulated upon Hsp70 inhibition were no longer VCP positive, disaggregation by VCP may allow Hsp70 to access aggregates of non-ubiquitylated Tau. This possibility is consistent with the reported ability of the Hsp70 system to disaggregate Tau aggregates in vitro[16]. Moreover, owing to the structural diversity of Tau filaments among different tauopathies[69], different forms of Tau aggregates may be more or less suitable for disaggregation by VCP or Hsp70, and degradation by proteasome or autophagy.

Support for the physiological relevance of VCP in antagonizing amyloid aggregation is provided by mutations in VCP that are associated with the deposition of ubiquitylated aggregates in neurodegenerative diseases such as vacuolar tauopathy and IBMPFD[21,29,61]. However, the vacuolar tauopathy-associated VCP mutation D395G[21] and the IBMPFD mutations A232E and R155H[27] did not detectably impair VCP-mediated Tau disaggregation in our cellular model. Whether the impairment of disaggregation in a neuronal model is more pronounced, remains to be tested. Although even a small inhibitory effect on disaggregation may contribute to aggregate pathology in neurons over decades, the disease mutations may alternatively affect other steps during aggregate formation, including aggregate seeding and Tau degradation in cooperation with the proteasome. Indeed, an increase in intracellular aggregation is observed when VCP is inhibited in recipient cells at the time of seeding[64] or when VCP D395G is expressed in the recipient cells[21]. Note that we introduced the mutant VCP proteins in cells containing preexistent Tau aggregates to exclude a potential role of VCP in the process of aggregate seeding. Other potential mechanisms by which VCP mutations may lead to the disease include perturbed autophagy[35], reduced clearance of damaged mitochondria[70], respiratory deficits, and reduced ATP production[71].

Our finding that clearance of Tau aggregates by VCP generates smaller seeding competent species as a byproduct (Fig. 7) provides a plausible explanation for how VCP can exert both neuroprotective and neurotoxic effects. While we demonstrate these effects in a cellular system, indeed, overexpression of a VCP homologue in a *Drosophila*

*melanogaster* model of polyglutamine protein aggregation hastened the degenerative phenotype[72]. Transcellular aggregate spreading has been recognized as a major driver of neurodegenerative disease progression[6,7,73], and the generation of seeding competent species may be an inevitable consequence of amyloid clearance mechanisms via disaggregation, not only by VCP but also by the Hsp70 system[16]. We note, however, that in contrast to inhibition of VCP, neither Hsp70 nor proteasome inhibition had a significant effect on the generation of seeding competent Tau species in our model, suggesting that their function is not directly coupled to seed production. Furthermore, as our understanding of cellular seed release and uptake is limited, how disaggregated Tau fragments are transferred among cells to induce further Tau aggregation remains to be addressed. Seeds may be released to the extracellular space by damaged cells or by exocytosis and taken up by neighboring cells via receptor-mediated endocytosis or via phagocytosis[7]. Additionally, Tau fibril fragments in the extracellular space may be recognized by receptors on microglial cells resulting in their activation and release of inflammatory molecules, thereby contributing to neuroinflammation observed in diseased brains[74].

VCP-mediated aggregate disassembly followed by proteasomal degradation provides an important alternative to autophagy as a mechanism for the elimination of terminally aggregated proteins. Based on our results, both activation and inhibition of this pathway may have beneficial effects dependent on the specific disease context. Non-human AAA + ATPases with augmented disaggregase activity are currently being developed with the aim to reverse pathogenic protein aggregation[75,76]. Boosting cellular aggregate clearance, perhaps in combination with proteasome activation[77], may offer a potential therapeutic strategy as long as the production of seeding competent species can be controlled.

## Methods
### Plasmids
The N1-TauRD (P301L/V337M)-EYFP and N1-FLTau (0N4R, P301L/V337M)-EYFP constructs were previously described[39,65]. To generate Tet-off TauRD (P301L/V337M) (RRID:Addgene_188572) and Tet-off Full

Length Tau (0N4R, P301L/V337M) (RRID:Addgene_188573) constructs without fluorescent tag, a stop codon was introduced in the N1-TauRD (P301L/V337M)-EYFP and N1-FLTau (0N4R, P301L/V337M)-EYFP plasmids after the Tau sequence using the Q5 site directed mutagenesis (SDM) kit (New England Biolabs). Tau fragments were subcloned into pcDNA3.1 by restriction digestion and further into pCW57.1-MAT2A all-in-one tet-off lentiviral backbone (a gift from David Sabatini (RRID:Addgene_100521)[78] by Gibson assembly. TauRD (P301L/V337M) construct contains a C-terminal myc-tag separated from TauRD by a 4 aa (GGSG) linker.

Wild type (WT) VCP (RRID:Addgene_23971), A232E VCP (RRID:Addgene_23973), R155H VCP (RRID:Addgene_23972) and E305Q/E578Q VCP (RRID:Addgene_23974) sequences were derived from plasmids described previously[35]. A C-terminal myc tag and stop codon was introduced using SDM followed by subcloning the VCP-myc fragments into pcDNA3.1. The D395G VCP construct was generated by introducing the D395G mutation in WT-VCP plasmid by SDM. All mutations were verified by sequencing. The plasmid expressing wild type firefly luciferase fused to EGFP (Fluc-GFP) was previously described[59].

Lentiviral packaging plasmid pVsVg was a gift from Dieter Edbauer. psPAX2 (RRID:Addgene_12260) and pMD2.G (RRID:Addgene_12259) also used for lentiviral production were gifts from Didier Trono. pFhSynW2 TauRD (P301L/V337M)-EYFP used for TauRD-EYFP expression in mouse primary neurons was previously described[65].

## Cell lines and cell culture

Cells expressing constitutive and tet-regulated TauRD-Y (TauRD-Y and Tet-TauRD-Y cell lines, respectively), FRET biosensor TauRD-TY, and FLTau-Y cells were previously described[39,65]. Tet-FLTau (RRID:CVCL_C1CR) and Tet-TauRD (RRID:CVCL_C1CS) cell lines were generated by transducing HEK293T (RRID:CVCL_0063) cells with 200 μL concentrated lentivirus in presence of 0.8 μg/mL Polybrene (Sigma). Transduced cells were selected with 10 μg/mL Blasticidin (Thermo) and thereafter sorted in 96 well-plates with a BD FACS Aria III (BD Biosciences) (Imaging Facility, MPI Biochemistry). Monoclonal cell lines stably expressing FLTau and TauRD were screened by immunofluorescence staining and immunoblotting followed by amplification.

All cell lines were cultured in Dulbecco's Modified Eagle Medium (Biochrom) supplemented with 10% FBS (Gibco), 2 mM L-glutamine (Gibco), 100 units/mL penicillin and 100 μg/mL streptomycin (Gibco), and non-essential amino acid cocktail (NEAA) (Gibco) and grown at 37 °C at 5% $CO_2$. TauRD-TY and FLTau-Y cells were maintained in presence of 200 μg/mL G418 (Gibco). HEK293 cells stably expressing Fluc-GFP were maintained in presence of 50 μg/mL hygromycin (Thermo).

## Generation of cell lines propagating Tau aggregates

Tau aggregation was induced by addition of TauRD aggregates as described previously[39]. Briefly, HEK293 cells expressing TauRD-Y were initially treated with fibrillar aggregates generated in vitro from purified TauRD and clones that displayed the ability to maintain TauRD-Y aggregates for multiple passages were selected. Aggregate-containing TauRD-Y* cells were lysed in Triton buffer (0.05% Triton X-100/PBS (Gibco) supplemented with protease inhibitor cocktail (Roche, EDTA-free) and benzonase (prepared in-house)) and kept on ice for 20 min. Cell lysate was centrifuged at 1000 x g for 5 min and the supernatant was collected. Protein concentration in cell lysates was determined using Bradford assay (Bio-Rad). A total of 30 μg of freshly prepared lysate was diluted in 100 μL Opti-MEM Reduced Serum Medium (Gibco). In a separate tube 4 μL Lipofectamine 2000 was diluted in 100 μL Opti-MEM and incubated at room temperature (RT) for 5 min. Contents of the tubes were gently mixed and incubated at RT for 20 min. The lysate-lipofectamine mixture was applied to naive cells expressing soluble TauRD-Y, plated at 150,000 cells/well in a 12-well plate. 24 h later, cells were transferred to a 6-well plate and 3 days later to 10 cm dishes (<200 cells per dish) for 8 days, until clearly visible colonies were observed. Colonies were screened for the presence of YFP-positive aggregates with an Axio Observer fluorescent microscope (Zeiss). Monoclonal cells displaying aggregate morphology similar to parental cells were amplified and frozen until use. TauRD, FLTau and FLTau-Y expressing cells were similarly seeded with cellular TauRD aggregates[39] and cultured for several days before experiments were performed with a polyclonal cell population.

## Lentivirus production

For primary neuron transduction: HEK293T cells (LentiX 293T cell line, Takara) for lentiviral packaging were expanded to 70-85% confluency in DMEM Glutamax (+4.5 g/L D-glucose, - pyruvate) supplemented with 10% FBS (Sigma), 1% G418 (Gibco), 1% NEAA (Thermo Fisher) and 1% HEPES (Biomol). Only low passage cells were used. For lentiviral production, a three-layered 525 cm² flask (Falcon) was seeded and cells were henceforth cultured in medium without G418. On the following day, cells were transfected with the expression plasmid pFhSynW2 (TauRD-Y), the packaging plasmid psPAX2 and the envelope plasmid pVsVg using TransIT-Lenti transfection reagent (Mirus). The transfection mix was incubated for 20 min at RT. The cell medium was exchanged in the meantime. 10 mL of transfection mix was added to the flask, followed by incubation overnight. The medium was exchanged on the following day. After 48–52 h, culture medium containing the viral particles was collected and centrifuged for 10 min at 1200 x g. The supernatant was filtered through 0.45 μm pore size filters using 50 mL syringes, and Lenti-X concentrator (Takara) was added. After an overnight incubation at 4 °C, samples were centrifuged at 1500 x g for 45 min at 4 °C, the supernatant was removed and the lentivirus pellet was resuspended in 150 μL TBS-5 buffer (50 mM Tris-HCl pH 7.8, 130 mM NaCl, 10 mM KCl, 5 mM $MgCl_2$). After aliquoting, lentivirus was stored at −80 °C. A detailed protocol is available at dx.doi.org/10.17504/protocols.io.6qpvr4xn3gmk/v1.

For HEK293T transduction: HEK293T cells (LentiX 293T cell line, Takara) were transfected in 10 cm dishes with packaging plasmid psPAX2, envelope plasmid pMD2.G and expression plasmids (pCW Tet-off FLTau and TauRD) using Lipofectamine 3000. Forty-eight hours later virus-containing media was harvested and centrifuged for 5 min at 1000 x g. Lenti-X concentrator was added to supernatant, incubated overnight at 4 °C and the following day centrifuged for 45 min at 1500 x g at 4 °C. The lentiviral pellet was resuspended in 1 mL PBS, aliquoted and stored at −80 °C. A detailed protocol is available at dx.doi.org/10.17504/protocols.io.kqdg39eo1g25/v1.

## Primary neuronal cultures

Primary cortical neurons were prepared from E15.5 CD-1 wild type mouse embryos. All experiments involving mice were performed in accordance with the relevant guidelines and regulations. Pregnant female mice were sacrificed by cervical dislocation. The uterus was removed from the abdominal cavity and placed into a 10 cm sterile Petri dish on ice containing dissection medium, consisting of Hanks' balanced salt solution (HBSS) supplemented with 0.01 M HEPES, 0.01 M $MgSO_4$ and 1% penicillin/streptomycin. Each embryo was isolated, heads were quickly cut, and brains were removed from the skull and immersed in ice-cold dissection medium. Cortical hemispheres were dissected, and meninges were removed. The cortices were collected in a 15 mL sterile tube and digested with 0.25% trypsin containing 1 mM ethylenediaminetetraacetic acid (EDTA) and 15 μL 0.1% DNAse I for 20 min at 37 °C. The digestion was stopped by removing the supernatant and washing the tissue twice with Neurobasal medium (Invitrogen) containing 5% FBS. The tissue was resuspended in 2 mL Neurobasal medium and triturated to achieve a single cell suspension. Cells were spun at 130 x g, the supernatant was removed, and the cell pellet was resuspended in Neurobasal medium with 2% B-27

supplement (Invitrogen), 1% L-glutamine (Invitrogen) and 1% penicillin/ streptomycin (Invitrogen). For immunofluorescence microscopy, neurons were cultured in 24-well plates on 13 mm coverslips coated with 1 mg/mL poly-D-lysine (Sigma) and 1 μg/mL laminin (Thermo Fisher Scientific) (100,000 neurons per well). For viability measurements, neurons were cultured in 96-well plates coated in the same way (18,000 neurons per well). Lentiviral transduction was performed at 10 days in vitro (DIV 10). Virus preparation was thawed and immediately added to freshly prepared neuronal culture medium. Neurons in 24-well plates received 1 μL of virus per well. Neurons in 12-well plates received 1.5 μL of virus per well. Neurons in 96-well plates received 0.15 μL of virus per well. A fifth of the medium from cultured neurons was removed and the equivalent volume of virus-containing medium was added. Three days after transduction (DIV 13), 2, 6, or 12 μg of HEK293 cell lysate containing TauRD-Y aggregates, mixed with fresh medium (one tenth of medium volume in the well), were added to the neuronal cultures in 96, 24, or 12-well plates, respectively. HEK293 cell lysate for neurons was prepared by brief sonication of aggregate-containing cells in PBS. Six days after transduction (DIV 16), neurons were treated with inhibitor or DMSO as control. A detailed protocol for the preparation of primary neuronal cultures is available at dx.doi.org/ 10.17504/protocols.io.ewov1ojwklr2/v1.

## Neuronal viability assay

Viability of transduced neurons was determined using Thiazolyl Blue Tetrazolium Bromide (MTT; Sigma-Aldrich). Seven days after transduction (DIV 17), cell medium was exchanged for 100 μL of fresh medium, followed by the addition of 20 μL of 5 mg/ml MTT/PBS and incubation for 4 h at 37 °C, 5% $CO_2$. Subsequently, 100 μL solubilizer solution (10% SDS, 45% dimethylformamide in water, pH 4.5) was added, and on the following day, absorbance was measured at 570 nm. Each condition was measured in triplicates per experiment and absorbance values were averaged for each experiment. The individual values for the 'Control-Seed' condition obtained for each of the three experiments were normalized by the mean of these values. The values of all other conditions were normalized by the new value of the 'Control-Seed' condition of the corresponding independent experiment.

## Plasmid and siRNA transfection

Plasmids were transfected with Lipofectamine 2000 (Thermo) after manufacturer's instructions in 12- or 6-well plates using 2 or 4 μg DNA. All siRNAs were obtained from Dharmacon as ON-TARGETplus SMART pools: VCP (L-008727-00-0005), Atg5 (M-004374-04-0005), Atg7 (L-020112-00-0005), PSMD11 (L-011367-01-0005), non-targeting control (D-001810-03-20). Cells were plated in 24-well plates in 500 μL antibiotic free DMEM. 2 μL of Dharmafect transfection reagent and 50-100 nM of siRNA were diluted each in 50 μL Opti-MEM and incubated at RT for 5 min. The contents of the tubes were mixed gently by pipetting and incubated further at RT for 15 min. Subsequently, the transfection mixture was added to the cells drop-wise. Twenty-four hours later cells were split and plated in 12- or 6-well plates and allowed to grow for up to 96 h before immunoblotting or immunofluorescent staining.

## Antibodies and chemicals

The following primary antibodies were used for immunoblotting or immunofluorescent staining: anti-VCP (Abcam Cat# ab11433, RRID:AB_298039), anti-VCP (Novus Biologicals Cat# NB 100-1558, RRID:AB_527461) (Fig. 2e, Supplementary Fig. 11a and Supplementary Fig. 6g), anti-GFP (Roche Cat# 11814460001, RRID:AB_390913), anti-ubiquitin Lys48-specific (Millipore Cat# 05-1307, RRID:AB_1587578), anti-ubiquitin Lys63-specific (Millipore Cat# 05-1308, RRID:AB_1587580), anti-ubiquitin (P4D1) (Santa Cruz Biotechnology Cat# sc-8017, RRID:AB_628423), anti-Tau (pS356) (GeneTex Cat# GTX50165,

RRID:AB_11167058), anti-phospho-Tau (S202, T205) (Thermo Fisher Scientific Cat# MN1020, RRID:AB_223647), anti-NPLOC4 (Sigma-Aldrich Cat# HPA021560, RRID:AB_1854586), anti-UFD1L (Abcam Cat# ab96648, RRID:AB_10678868), anti-ubiquitin FK2 (Millipore Cat# 04-263, RRID:AB_612093), anti-Tau (Tau-5) (Thermo Fisher Scientific Cat# MA5-12808, RRID:AB_10980631), anti-human Tau/Repeat Domain (2B11) (Tecan (IBL) Cat# JP10237, RRID:AB_2341273), anti-LC3B (Sigma-Aldrich Cat# L7543, RRID:AB_796155), anti-Atg5 (Cell Signaling Technology Cat# 2630, RRID:AB_2062340), anti-Atg7 (Cell Signaling Technology Cat# 8558, RRID:AB_10831194), anti-PSMD11 (Proteintech Cat# 14786-1-AP, RRID:AB_2268979), anti-myc (in house, DSHB Cat# 9E 10, RRID:AB_2266850), anti-GAPDH (Millipore Cat# MAB374, RRID:AB_2107445), anti-Tubulin (Sigma-Aldrich Cat# T6199, RRID:AB_477583).

The following secondary antibodies were used: Cy5-conjugated anti-mouse (Thermo Fisher Scientific Cat# A10524, RRID:AB_2534033), Cy-5 conjugated anti-rabbit (Thermo Fisher Scientific Cat# A10523, RRID:AB_2534032), Alexa Fluor 647 AffiniPure anti-mouse (Jackson ImmunoResearch Labs Cat# 715-605-151, RRID:AB_2340863), DyLight 488 anti-mouse (Thermo Fisher Scientific Cat# SA5-10166, RRID:AB_2556746), anti-mouse IgG peroxidase conjugate (Sigma-Aldrich Cat# A4416, RRID:AB_258167) or anti-rabbit peroxidase conjugate (Sigma-Aldrich Cat# A9169, RRID:AB_258434), IRDye 680RD anti-mouse (LI-COR Biosciences Cat# 926-68070, RRID:AB_10956588), IRDye 800CW anti-rabbit (LI-COR Biosciences Cat# 926-32211, RRID:AB_621843).

The following chemicals were used: Cycloheximide (Sigma), doxycycline (Sigma), 3-methyladenine (Invivogen), bafilomycin A1 (Invivogen), epoxomicin (Cayman Chemical), NMS-873 (Sigma), CB-5083 (Cayman Chemical), VER-155008 (Sigma), MLN7243 (Chemietek). Solutions in DMSO were stored at −20 °C. 3-Methyadenine was dissolved in $H_2O$ after manufacturer's instructions and applied immediately to cells.

## Immunofluorescence staining

HEK293 cells were grown on poly-L-lysine (NeuVitro) coated glass coverslips for 24–48 h in 12-well plates before any treatment. At the end of the experiment, media was aspirated and cells were directly fixed in 4% formaldehyde (w/v) (Thermo, Methanol-free) in PBS for 10 min at RT, washed once with PBS and permeabilized in 0.1% Triton X-100/PBS for 5 min. Samples were blocked using 5% low-fat dry milk dissolved in 0.1% Triton X-100/PBS for 1 h at RT, followed by incubation with primary antibodies in blocking solution and fluorescently labelled secondary antibodies in PBS. Nuclei were counterstained with DAPI. For amyloid staining, after fixation and permeabilization, cells were incubated with Amylo-Glo (Biosensis TR-300-AG) at a dilution of 1:200/PBS with gentle shaking followed by washing twice with PBS. Cells were not counterstained with DAPI. Coverslips were mounted in fluorescent mounting medium (Dako) on glass slides and stored at 4 °C until imaging.

**Primary neurons.** Primary neurons were fixed at DIV 17 with 4% paraformaldehyde (Santa Cruz) (PFA)/PBS for 15 min; remaining free aldehyde groups of PFA were blocked with 50 mM ammonium chloride/PBS for 10 min at RT. Cells were rinsed once with PBS and permeabilized with 0.25% Triton X-100/PBS for 5 min. After washing with PBS, blocking solution consisting of 2% BSA (w/v) (Roth) and 4% donkey serum (v/v) (Jackson ImmunoResearch Laboratories) in PBS was added for 30 min at RT. Coverslips were transferred to a light protected humid chamber and incubated with primary antibodies diluted in blocking solution for 1 h. Cells were washed with PBS and incubated with secondary antibody diluted in blocking solution for 30 min and counterstained with DAPI. Coverslips were mounted using Prolong Glass fluorescence mounting medium (Invitrogen).

## Mice

All experiments involving mice were performed in accordance with the relevant guidelines and regulations of the Government of Upper Bavaria (Germany). Mice were maintained in a specific pathogen-free animal facility with ad libitum access to food and water. E15.5 CD-1 wild type mouse embryos were used for preparation of neuronal cultures. rTg4510 mice[5] were obtained by crossing tetO-tauP301L mice (JaxLabs, Stock number 015815) to the CamKIIα-tTA line[79] (JaxLabs, Stock number 003010) and maintained on a C57BL/6 genetic background. Animals of either sex were used for the experiments. Littermates were used as controls.

## Immunohistochemistry

Brain sections from 3 mice were analyzed; one 16-month-old and two 4-month old rTg4510 and 4-month-old control mice. Mice were deeply anesthetized with 1.6% Ketamine/0.08% Xylazine and transcardially perfused with PBS followed by 4% PFA in PBS. Brains were dissected out of the skull and post-fixed in 4% PFA in PBS overnight. Fixed tissue was embedded in agarose and sectioned into 40 μm thick sections using a vibratome (VT1000S, Leica). Sections were permeabilized with 0.5% Triton X-100. After washing with PBS, sections were incubated in blocking solution consisting of 0.2% BSA (w/v), 5% donkey serum (v/v), 0.2% lysine (w/v) (Sigma-Aldrich), 0.2% glycine (w/v) (Sigma-Aldrich) in PBS for 30 min at RT. Sections were incubated with primary antibodies diluted in 0.3% Triton X-100 (v/v), 2% BSA (w/v) in PBS overnight at 4 °C. Sections were washed in PBS and incubated with secondary antibodies diluted in 0.3% Triton X-100, 3% donkey serum (v/v) for 2 h at RT, with 0.5 μg/ml DAPI added to stain the nuclei. Sections were mounted on Menzer glass slides using Prolong Glass fluorescence mounting medium. A detailed protocol is available at dx.doi.org/10.17504/protocols.io.x54v9den4g3e/v1.

## Image acquisition (Microscopy)

Images were acquired with a Zeiss LSM 780 (software ZEN 2011 SP7 (black) v14.0.2.201), Leica SP8 FALCON (software LAS X v3.5.7.23225) (Imaging Facility, MPI Biochemistry) or a Leica TCS SP8 Laser-scanning confocal microscope (software LAS X v3.5.7.23225) (Imaging Facility, MPI Biological Intelligence) and analyzed using FIJI/ImageJ software v1.49 s (RRID:SCR_002285; http://fiji.sc). For multifluorescent imaging, samples stained with individual fluorophores were used to correct emission bandwidths and exposure settings to minimize spectral crossover.

## Quantification of aggregates/cell and average size

Confocal z-stacks were used to create a maximum intensity projection (MIP) using the image acquisition software ZEN 2011 SP7 (black) v14.0.2.201 (Zeiss) (RRID:SCR_018163). MIPs were further segmented to define aggregate foci by thresholding. Aggregate number and size were computed by the Analyze Particle function (Size: 0-infinity). Cell numbers were determined by counting DAPI-stained nuclei with the Cell Counter plugin (https://imagej.nih.gov/ij/plugins/cell-counter.html). Experiments were performed at least 3 times in biologically independent repeats. For neuronal aggregates, neuronal cytoplasm area was calculated by manually selecting a region of interest (ROI) around the soma of the neuron and utilizing the Analyze feature. Aggregate foci were identified by thresholding the MIP images and aggregate size (area), within the previously selected ROI, was calculated by the Analyze Particle function. The percentage of total neuron area occupied by aggregate was the quotient of the division between aggregate area and neuronal cytoplasmic area: (Aggregate area/Cytoplasm area) x 100. Sixty individual neurons were imaged per condition, in three biologically independent replicates. Raw data of these experiments are provided in the Source Data file.

## mRNA quantification

Total RNA was isolated using the RNeasy Mini Kit (Qiagen) and reverse transcribed with iScript™ cDNA Synthesis Kit (Biorad) according to manufacturers' instructions. Quantitative real-time PCR was performed with PowerUp™ SYBR™ Green Master Mix (Applied Biosystems) with a StepOnePlus Real-Time PCR System (Applied Biosystems). CT values were measured and fold changes were calculated by the ΔΔC(T) method[80] using the RPS18 gene as reference. The following primers were used: RPS18 forward 5′-TGTGGTGTTGAGGAAAGCA-3′ and reverse 5′- CTTCAGTCGCTCCAGGTCTT-3′; Tau forward: 5′-AGCAACGTCCAGTCCAAGTG-3′ and reverse: 5′-CCTTGCTCAGGTCAACTGGT-3′. Raw data of this experiment is provided in the Source Data file.

## Correlative light and electron microscopy (CLEM), cryo-focused ion beam (FIB) and cryo-Electron Tomography

A total of $2 \times 10^4$ TauRD-Y* cells or $1 \times 10^5$ neurons were seeded on EM grids (R2/1, Au 200 mesh grid, Quantifoil Micro Tools) in a 35 mm dish or 24-well plate and cultured for 24 h or transduced with lentivirus and treated with aggregate-containing cell lysate as described earlier in section 'Primary neuronal cultures'. The grids were blotted for 10 s using filter paper and vitrified by plunge freezing into a liquid ethane/propane mixture with a manual plunger. CLEM, cryo-FIB and tomographic data collection were performed as described in detail before[81]. In brief, EM grids were mounted onto modified Autogrid sample carriers[82] and then transferred onto the cryo-stage of a CorrSight microscope (FEI) for cryo-light microscopy. Images of the samples and TauRD-Y signal were acquired with MAPS software v2.1 (FEI) in transmitted light and confocal mode using a 5x and 20x lens, respectively. The samples were then transferred into a dual-beam (FIB/SEM) microscope (Quanta 3D FEG, FEI) using a cryo-transfer system (PP3000T, Quorum). Cryo-light microscope and SEM images were correlated with MAPS software. Lamellas (final thickness, 100–200 nm) were prepared using a Ga$^{2+}$ ion beam at 30 kV in the regions of the TauRD-Y fluorescence signal. In case of TauRD-Y* cells, an additional layer of platinum was sputter-coated (10 mA, 5 s) on the grids to improve conductivity of the lamellas.

The grids were then transferred to a Titan Krios transmission electron microscope (FEI) for tomographic data collection. For the whole procedure, samples were kept at a constant temperature of −180 °C. Tomographic tilt series were recorded with a Gatan K2 Summit direct detector in counting mode. A GIF-quantum energy filter was used with a slit width of 20 eV to remove inelastically scattered electrons. Tilt series were collected from −50° to +70° with an increment of 2° and total dose of 110 e⁻/Å$^2$ using SerialEM software[83] v3.7.0 (RRID:SCR_017293; http://bio3d.colorado.edu/SerialEM/) at a nominal magnification of 33,000x, resulting in a pixel size of 4.21 Å for TauRD-Y* cells and at a nominal magnification of 42,000x, resulting in a pixel size of 3.52 Å for primary neurons. In case of TauRD-Y* cells, a Volta phase plate was used together with a defocus of −0.5 μm for contrast improvement[84].

For image processing of TauRD-Y* cell tomograms, frames were aligned during data collection using in-house software K2align based on previous work[85]. For tomograms of primary neurons, frames were aligned using the software Motioncor2 v1.2.1 (RRID:SCR_016499) and Tomoman (https://github.com/williamnwan/TOMOMAN). The IMOD software package[86] v4.9.0 (RRID:SCR_003297; http://bio3d.colorado.edu/imod) was used for tomogram reconstruction. The tilt series were first aligned using fiducial-less patch tracking, and tomograms were then reconstructed by weighted back projection of the resulting aligned images.

For segmentation, tomograms were rescaled with a binning factor of four and in case of the primary neurons tomograms filtered with a deconvolution filter (https://github.com/dtegunov/tom_deconv). Tau filaments were traced with XTracing Module in Amira v6.2 (Thermo

Fisher Scientific; RRID:SCR_007353) using a short cylinder as a template[87]. The membranes were first segmented automatically with TomoSegMemTV[88] using tensor voting, and then manually optimized in Amira. Segmentations were used only to visualize representative fibrillar Tau structures.

## Immunoblotting

Cells were lysed in RIPA lysis and extraction buffer (Thermo) supplemented with protease inhibitor cocktail and benzonase for 30 min on ice with intermittent vortexing. Protein concentration in total cell lysates was determined using Bradford assay (Bio-Rad) and normalized in all samples before adding 2x SDS sample buffer. Samples were denatured by boiling at 95 °C for 5 min. Proteins were resolved on NuPAGE 4-12% gradient gels (Thermo) with MES or MOPS (Thermo) running buffer at 200 V for 45 min. Proteins were transferred to nitrocellulose or PVDF membranes (Roche) in tris-glycine buffer at 110 V for 1 h. Membranes were washed once in TBS-T and blocked in 5% low-fat dry milk dissolved in TBS-T for 1 h at RT. Subsequently, blots were washed 3 times with TBS-T and probed with primary and secondary antibodies. Chemiluminescence was developed using HRP substrate (Luminata Classico, Merk) and detected on a LAS 4000 (Fuji) or ImageQuant800 (Amersham) imager with control software v1.2.0. AIDA image software v4.27.039 (Elysia Raytest) was used to quantify intensity of protein bands. Images were processed using ImageJ v1.49 s or Adobe Photoshop CC 2018. Uncropped scans of blots in main figures and supplementary figures are provided in Source Data file and at the end of Supplementary Information file, respectively.

## Interactome analysis by mass spectrometry

**SILAC labelling of cells and TauRD-Y immunoprecipitation.** Interactome analyses were performed using a stable isotope labelling by amino acids in cell culture (SILAC)-based quantitative proteomics approach[89]. Frozen TauRD-Y and TauRD-Y* cells were thawed in arginine lysine deficient SILAC media (PAA) containing light (L) ($Arg_0$, $Lys_0$, Sigma) and heavy (H) ($Arg_{10}$, $Lys_8$, Silantes) amino acid isotopes, respectively, and supplemented with 10% dialyzed FCS (PAA), 2 mM L-glutamine (Gibco), 100 units/mL penicillin and 100 µg/mL streptomycin (Gibco), and non-essential amino acid cocktail (Gibco). A third cell line, not part of this study but included in the PRIDE entry PXD023400, was simultaneously expanded in SILAC medium supplemented with medium (M) ($Arg_6$, $Lys_4$, Silantes) amino acid isotopes, and was processed and analyzed together with TauRD-Y and TauRD-Y* samples. The data from this cell line was not used in the study. Cells were passaged for a minimum of two weeks to allow efficient incorporation of amino acid isotopes into the cellular proteome. Cells from a 10 cm dish were washed in PBS, lysed by gentle pipetting in 400 µL ice-cold lysis buffer (1% Triton X-100/PBS supplemented with protease inhibitor cocktail and benzonase). Lysates were sonicated briefly and centrifuged at 2000 x g for 5 min at 4 °C. A total of 300 µL of the supernatant was removed and protein concentration was determined using Bradford assay (Bio-Rad). A total of 50 µL anti-GFP beads (µMACS GFP Isolation kit, Miltenyi Biotech) were added to 500 µg total protein diluted in a total volume of 800 µL lysis buffer. Lysates were incubated for 1 h at 4 °C with end-over-end rotation at 10 rpm. Micro-Columns (Miltenyi Biotech) were placed in the magnetic field of a µMACS Separator (Miltenyi Biotech) and equilibrated with 250 µL lysis buffer before lysates were applied. Columns were washed 4 times with 1 mL cold Triton buffer and 2 times with 1 mL PBS followed by elution in 70 µL preheated 1x SDS sample buffer without bromophenol blue.

**MS sample processing.** Twenty microlitres of sample from each of the H, M, and L eluates were mixed and processed by the filter-aided sample preparation (FASP) method as previously described[90]. Samples were loaded in a 30 kDa centrifugation device and washed 3 times with

200 µL freshly prepared urea buffer (UB) (8 M urea, 0.1 M Tris pH 8.5). Reduction and alkylation was performed sequentially using 10 mM DTT and 50 mM iodoacetamide in UB, respectively. Samples were washed 2 times with 200 µL 50 mM ammonium bicarbonate ($NH_4HCO_3$) to remove urea before an over-night trypsin treatment. Peptides were recovered in 40 µL $NH_4HCO_3$, acidified with 12 µL of a 25% TFA solution and dried in a vacuum concentrator. The peptides were further fractionated using home-made SAX columns in 200 µL microtips by stacking 2 punch-outs of Empore High Performance Extraction Disk (Anion-SR) material. Peptides were sequentially eluted with 6 different Britton & Robinson buffers (BURB) of decreasing pH (pH 11, 8, 6, 5, 4, 3) and acidified to 1% TFA. The last elution step was with MeOH/water (1:1)/1% formic acid. The fractionated peptides were desalted with home-made micro-columns containing C18 Empore disks and eluted with 70% ACN 1% formic acid followed by drying in a vacuum concentrator. The samples were stored at −20 °C until analysis.

**LC-MS.** The desalted peptides were dissolved in 5 µL of 5% formic acid, sonicated in an ultrasonic bath, centrifuged and transferred to MS autosampler vials. Samples were analyzed on an Easy nLC-1000 nanoHPLC system (Thermo) coupled to a Q-Exactive Orbitrap mass spectrometer (Thermo). Peptides were separated on home-made spray-columns (ID 75 µm, 20 cm long, 8 µm tip opening, New-Objective) packed with 1.9 µm C18 particles (Reprosil-Pur C18-AQ, Dr Maisch GmbH) using a stepwise 115 min gradient between buffer A (0.2% formic acid in water) and buffer B (0.2% formic acid in acetonitrile). Samples were loaded on the column by the nanoHPLC autosampler at a flow rate of 0.5 µL per min. No trap column was used. The HPLC flow rate was set to 0.25 µL per min during analysis. MS/MS analysis was performed with standard settings using cycles of 1 high resolution (70000 FWHM setting) MS scan followed by MS/MS scans (resolution 17500 FWHM setting) of the 10 most intense ions with charge states of two or higher. Details of standard settings include Capillary temperature: 250 °C; Spray Voltage: 1500 V; MS Resolution: 70,000; MS/MS Resolution: 17,500, MS AGC target: 3e6; MS/MS AGC target: 2e5; Maximum MS IT: 120 ms; MS Scan range: 300 to 1750 m/z; MS/MS Scan range: 200 to 2000 m/z; Gas phase fragmentation method: HCD; Fragmentation relative energy level: 25; TopN (MS/MS precursors): 10; Isolation window 3.0 m/z; Underfill ratio: 0.1 %; MS/MS Intensity Treshold: 0.1%; MS and MS/MS data type: Profile; Dynamic Exclusion: 20 s; MS/MS Charge Exclusion: unassigned, singly charged; Peptide match: preferred; Exclude isotopes: on.

**MS data analysis.** Protein identification and SILAC based quantitation was performed using MaxQuant v1.5.4.1 (RRID:SCR_014485; https://www.maxquant.org) using default settings. The following processing parameters were used for identification and quantitation with MaxQuant- Enzymatic specificity: Trypsin (K,R) including cleavages next to Proline (at K,R); Missed cleavage sites: 2; Fixed Modification: Cystein Carbamidomethylation; Variable Modifications: Oxidation of Methionine, Acetylation of Protein N-termini; Minimum peptide length: 7 amino acids; Max peptide mass: 4600 Da; Protein FDR: 0.01; First search peptide MS mass tolerance (before re-calibration): 20 ppm; MS/MS mass tolerance: 20 ppm. The human sequences of UNIPROT v2019-03-12 (https://www.uniprot.org/proteomes/UP000005640) were used as a database for protein identification. MaxQuant used a decoy version of the specified UNIPROT database to adjust the false discovery rates for proteins and peptides below 1%. We used normalized MaxQuant ratios for enrichment analyses to correct for uneven total protein amounts in the SILAC-labeling states. Proteins quantified in at least two experiments with normalized H/L ratios ≥2 were considered as interactors of TauRD-Y in TauRD-Y* cells. Detailed statistical analysis of this cutoff criterion was not performed in this study, however, a false

positive rate below 1% with a less stringent cutoff of H/L > 1.66 using control experiments was previously determined[91]. Volcano plot including p-values of interactors was generated from H and L ratios using Perseus v1.6.2.3 (RRID:SCR_015753; https://www.maxquant.org/perseus).

### Biochemical detection of aggregated Tau

Cells were lysed for 30 min on ice in lysis buffer followed by brief sonication or 1 h in RIPA buffer. Lysates were centrifuged at 2000 or 1000 x *g* for 5 min. The supernatant was carefully removed and protein concentration was normalized across all samples. Lysates were then used for solubility or filter trap assays. Lysates were centrifuged at 186,000 x *g* for 1 h at 4 °C. Supernatant was removed and the pellet was washed with 200 μL PBS and centrifuged again for 30 min. Pellets were disintegrated in PBS by pipetting and boiled in 1x SDS sample buffer. Filter trap assays were performed with 200 μg total protein diluted in 200 μL lysis buffer. A cellulose acetate membrane (0.2 μm pore size, GE Healthcare) was pre-equilibrated in 0.1% SDS and affixed to the filter trap apparatus (PR648 Slot Blot Blotting Manifold, Hoefer). Samples were loaded and allowed to completely pass through the filter under a vacuum. Wells were washed 3 times with 200 μL 0.1% SDS/H$_2$O followed by standard immunoblotting of the membrane. A detailed protocol is available at dx.doi.org/10.17504/protocols.io.n92ldpbx8l5b/v1.

### Detection of Tau ubiquitylation

Cells were lysed as described in section Immunoblotting, with the addition of 20 mM N-ethylmaleimide followed by brief sonication and centrifugation at 2000 x *g* for 5 min. Protein concentration was determined using Bradford assay (Bio-Rad). A total of 50 μL anti-GFP beads were added to 1 mg total protein diluted in a total volume of 600 μL RIPA buffer. Lysates were incubated for 1 h at 4 °C with end over end rotation. Cell lysates were applied to μ-columns equilibrated with 250 μL RIPA buffer. Columns were washed four times with 1 mL 0.1% SDS/PBS. Bound proteins were eluted by applying 50 μL pre-heated (95 °C) 1x SDS sample buffer. Input and eluates were resolved on NuPAGE 4–12% gradient gels in MOPS running buffer and transferred to nitrocellulose membranes. Membranes were probed with antibodies against GFP or ubiquitin-K48. A detailed protocol is available at dx.doi.org/10.17504/protocols.io.3byl4je4rlo5/v1.

### Native-PAGE analysis

Tet-TauRD-Y* cells were plated in 12-well plates and transfected with VCP variants using Lipofectamine 2000 for 2 days. Cells were then lysed in 50 μL 0.5% TritonX-100/PBS supplemented with protease inhibitor cocktail and benzonase for 1 h on ice. Lysates were centrifuged at 10,000 x *g* for 2 min and supernatant was collected. Protein concentration in the supernatant was determined using Bradford assay and normalized in all samples before adding 2x native sample buffer (40 % glycerol, 240 mM Tris pH 6.8, 0.04 % bromophenol blue). Samples were analyzed on Novex Value 4 to 12% Tris-glycine gels (Thermo) using 20 mM Tris 200 mM Glycine buffer at pH 8.4. Proteins were transferred to nitrocellulose membrane in Tris-glycine buffer, blocked in 5% low-fat dry milk and co-incubated with primary followed by fluorescent secondary antibodies. A fluorescent signal was detected on an Odyssey Fc imager (LI-COR). A detailed protocol is available at dx.doi.org/10.17504/protocols.io.36wgqjrzyvk5/v1.

### TauRD-Y seeding assay

TauRD-Y* cells were treated with 2 μM NMS, 10 μM VER, or 50 nM epoxomicin for 24 h or 0.5 μM MLN for 12 h, or with DMSO as control and lysed on ice in Triton buffer supplemented with protease inhibitor cocktail and benzonase for 20 min. The amount of TauRD-Y across the samples was normalized by quantifying TauRD-Y by immunoblotting using anti-GFP antibody and anti-GAPDH antibody as loading control. Lysates containing equal amounts of TauRD-Y were combined with Opti-MEM and Lipofectamine 3000, incubated for 20 min at RT, and added to FRET biosensor cells. 24 h later, cells were harvested with trypsin, washed with PBS, and analyzed on an Attune NxT flow cytometer (Imaging Facility, MPI Biochemistry). mTurquoise2 and FRET fluorescence signals were measured by exciting cells with a 405 nm laser and collecting fluorescent signal with 440/50 and 530/30 filters, respectively. To measure the YFP fluorescence signal, cells were excited with a 488 nm laser and emission was collected with a 530/30 filter. For each sample 50,000 single cells were evaluated. Data were processed using FlowJo v10.7.1 software (FlowJo LLC) (RRID:SCR_008520; www.flowjo.com/solutions/flowjo). After gating single cells, an additional gate was introduced to exclude cells that generate a false-positive signal in the FRET channel due to excitation at 405 nm[92]. A FRET positive gate was drawn by plotting the FRET fluorescence signal versus the mTurquoise2 fluorescence signal using unseeded cells as reference (Supplementary Fig. 13). A detailed protocol is available at dx.doi.org/10.17504/protocols.io.x54v9jnrzg3e/v1.

### Size exclusion chromatography of cell lysates

TauRD-Y* cells that had been treated for 24 h with DMSO or 2 μM NMS were analyzed. Untreated TauRD-Y cells were analyzed as control. Cells were lysed as described in the section Seeding assay. Lysates were clarified by centrifugation at 1000 x *g* for 5 min at 4 °C and filtered with a PVDF 0.22 μm filter (Millex). The total protein amount of the lysates was determined by Bradford assay (Bio-Rad). Three milligrams of total protein were loaded on a Superose 6 HR10/30 (GE Healthcare) column equilibrated with PBS. The individual fractions separated by size exclusion chromatography were analyzed and quantified by immunoblotting using anti-GFP antibody. TauRD-Y species were detected in the void volume (HMW) and low molecular weight (LMW) fractions. Corresponding fractions were pooled and analyzed by immunoblotting using anti-GFP antibody. Seeding experiments were performed as described above, using 0.5 ng TauRD-Y from HMW and LMW fractions. A detailed protocol is available at dx.doi.org/10.17504/protocols.io.4r3l27mjpg1y/v1.

### Statistical analysis

Statistical analysis was performed in Excel, Origin 2019b (RRID:SCR_014212) or GraphPad Prism 7 (RRID:SCR_002798) on data acquired from at least three independent experiments. Matched samples were compared using two-tailed Student's paired *t*-test. For multiple comparisons, one-way ANOVA followed by a Tukey post hoc test was used.

### Reporting summary

Further information on research design is available in the Nature Portfolio Reporting Summary linked to this article.

## Data availability

All data supporting the findings of this study are included in the manuscript and the Supplemental Information. Source data are provided in this paper. Additional data are available from the corresponding author upon reasonable request. The mass spectrometry proteomics data associated with Fig. 2a have been deposited to the ProteomeXchange Consortium via the PRIDE[93] partner repository (https://www.ebi.ac.uk/pride/archive/) with the dataset identifier PXD023400. This PRIDE entry additionally contains analyses that are not a part of this study.

The tomograms shown in Fig. 1d and Fig. 3c are available in the EMDB https://www.ebi.ac.uk/emdb/ through the following information:

EMD ID: EMD-13739 (Fig. 1d) and EMD-13740 (Fig. 3c) Source data are provided with this paper.

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

## Acknowledgements

We thank Ralf Zenke, Markus Oster, Giovanni Cardone and Martin Spitaler from the MPIB Imaging Facility for assistance with confocal microscopy, image analysis and flow cytometry, and Albert Ries for assistance with mass spectrometry. We acknowledge Gopal Jayaraj for generation of the Fluc-GFP cell line and Alonso Izzat Carvajal Alvarez for help with native PAGE analysis. We thank Sonja Blumenstock for managing the rTg4510 mouse colony and facilitating the access to the necessary tissue. The research leading to these results has received funding from the European Commission under Grant FP7 GA ERC- 2012-SyG_318987–ToPAG (I.S., R.K., P.Y.-C., I.D., R.F.-B., F.U.H. and M.S.H.), Marie Skłodowska-Curie grant (agreement no. 749370) (S.G.), the Deutsche Forschungsgemeinschaft (DFG, German Research Foundation) under Germany's Excellence Strategy within the framework of the Munich Cluster for Systems Neurology (EXC 2145 SyNergy – ID 390857198) (V.A.T., F.U.H. and M.S.H.) and MBExC (EXC 2067/1 — ID 390729940) (R.F.-B.), as well as by the joint efforts of The Michael J. Fox Foundation for Parkinson's Research (MJFF) and the Aligning Science Across Parkinson's (ASAP) initiative. MJFF administers the grant ASAP-000282 on behalf of ASAP and itself. M.S.H. acknowledges the funding from Alzheimer Nederland (Grant Number WE.03-2020-12).

## Author contributions

I.S. designed and performed most experiments. P.Y. performed seeding experiments. M.D.P. performed neuronal cultures, rTg4510 immuno-histochemistry and analyses. Q.G. and V.A.T. carried out cryo-electron tomography of Tau aggregates in TauRD-Y* cells and primary neurons, respectively. R.K. performed mass spectrometry analysis. S.G. helped with initial experiments and quantified inclusion size. H.H. performed mRNA analysis. I.D. supervised experiments with neuronal cultures and mouse immunohistochemistry. R.F.B. and W.B. supervised cryo-electron tomography experiments. D.W.S. and M.I.D. provided cell lines, protocols and contributed to the interactome analysis. F.U.H. and M.S.H. initiated and supervised the project and wrote the manuscript with input from I.S. and the other authors.

## Funding

## Competing interests

The authors declare no competing interests.

## Additional information

¹Department of Cellular Biochemistry, Max Planck Institute of Biochemistry, Am Klopferspitz 18, 82152 Martinsried, Germany. ²Aligning Science Across Parkinson's (ASAP) Collaborative Research Network, Chevy Chase, MD, USA. ³Molecular Neurodegeneration Group, Max Planck Institute for Biological Intelligence, 82152 Martinsried, Germany. ⁴Department of Molecules – Signaling – Development, Max Planck Institute for Biological Intelligence, Am Klopferspitz 18, 82152 Martinsried, Germany. ⁵Center for Anatomy, Faculty of Medicine and University Hospital Cologne, University of Cologne, 50931 Cologne, Germany. ⁶Department of Structural Molecular Biology, Max Planck Institute of Biochemistry, Am Klopferspitz 18, 82152 Martinsried, Germany. ⁷Munich Cluster for Systems Neurology (SyNergy), Munich, Germany. ⁸Institute of Neuropathology, University Medical Center Göttingen, 37099 Göttingen, Germany. ⁹Cluster of Excellence "Multiscale Bioimaging: from Molecular Machines to Networks of Excitable Cells" (MBExC), University of Göttingen, Göttingen, Germany. ¹⁰Center for Alzheimer's and Neurodegenerative Diseases, Peter O'Donnell Jr. Brain Institute, University of Texas Southwestern Medical Center, Dallas 75390 TX, USA. ¹¹School of Medicine and Health Sciences, Carl von Ossietzky University Oldenburg, Oldenburg, Germany. ¹²Department of Biomedical Sciences of Cells and Systems, University Medical Center Groningen, University of Groningen, Antonius Deusinglaan, 1, 9713 AV Groningen, The Netherlands. ¹³Present address: State Key Laboratory of Protein and Plant Gene Research, School of Life Sciences and Peking-Tsinghua Center for Life Sciences, Peking University, Beijing 100871, China. ¹⁴Present address: Department of Chemical and Biological Engineering, Princeton University, Princeton, NJ 08544, USA. ¹⁵Present address: Boehringer Ingelheim International GmbH, 55216 Ingelheim, Germany. ¹⁶Present address: ViraTherapeutics GmbH, 6063 Rum, Austria. ✉e-mail: uhartl@biochem.mpg.de; m.s.hipp@umcg.nl

