## [Peer Review File · Nature Communications]

REVIEWER COMMENTS

Reviewer #1 (Remarks to the Author):

In this manuscript, Saha et al. carry out experiments to demonstrate that the AAA+ chaperone VCP (p97) binds ubiquitylated Tau fibrils and disaggregates them. They demonstrate that inhibition of VCP activity stabilizes large Tau aggregates; however, this also results in the reduction in the amount of Tau species competent of prion like aggregate seeding in recipient cells. Therefore, the authors conclude that disaggregation by VCP generates seeding-active Tau as byproduct. Proper aggregate clearance requires the functions of Hsp70 and of the UPS system.

General impression

This is a very detailed study that presents a lot of data using multiple systems and multiple approaches to support the claims. However, it is mainly based on phenotypic observations. The authors have not done any of the needed in vitro experiments to show that isolated Tau fibrils can be disassembled by purified VCP. Furthermore, the role of Hsp70 is not well investigated and seems to be a side note in the manuscript.

Specific comments

1. The authors should quantify the inclusions/cell for Fig. 2d .
2. No filter-trap analysis done for CB-5083?
3. Filter-trap analysis of siVCP in Supplementary Fig. 4f are not very convincing, perhaps due to low efficiency of siRNA-mediated knockdown.
4. Line 171-172 vs line 203: Is the TauRD cell line tagged with myc or untagged? They are referred to differently in the two places indicated. If myc-tagged, naming conventions should be kept consistent, and should be renamed TauRD-M instead.
5. Claims of EQ/EQ mutants effects on TauRD-Y aggregates from lines 261-264 is not backed by actual data.
6. Is there a reason that the AE and RH mutants are not analyzed for whether they stabilize aggregates as the DG and EQ/EQ mutants in Fig. 5d? AE seems to colocalize nicely with aggregates in Fig. 5b.

Reviewer #2 (Remarks to the Author):

Thank you for writing a very comprehensive paper

A few general comments:
Lovely microscopy images!

Perhaps add into the title: "The AAA+ chaperone VCP disaggregates Tau fibrils and generates aggregate seeds " that this was a laboratory model

Please write out all abbreviations in the abstract for general readers who are not familiar with the

specific field: eg. valosin-containing protein (VCP)

In the abstract you state:

These findings identify VCP as a core component of the machinery for the removal of neurodegenerative disease aggregates and suggest that its activity can be associated with enhanced aggregate spreading in tauopathies.

Please add to your discussion how you relate your findings in the lab to what happens during neuro-inflammation in AD. Perhaps a figure could be added to the explanation.

VCP-mediated aggregate disassembly followed by proteasomal degradation provides an important alternative to autophagy as a mechanism for the elimination of terminally aggregated proteins.

Please add a diagram to explain autophagy and compare it to your VCP-mediated pathways and findings.

Please relate your findings directly to clinical application in AD. How do you foresee this research may impact clinical interventions?

Please elaborate or discuss in the conclusion if this work could be translated to other neuro-inflammatory diseases too? Parkinson's Disease?

Reviewer #3 (Remarks to the Author):

Saha et al. reported VCP disaggregates tau fibrils, however, it generates seeding-active tau as byproduct at the same time. These results suggest that VCP may have both neuroprotective and neurotoxic effects, although VCP mutations have been associated with aggregate deposition disorders such as vacuolar tauopathy and IBMPFD. Boosting cellular aggregate clearance, perhaps in combination with proteasome activation, may offer a potential therapeutic strategy as long as the production of seeding competent species can be controlled. This is an interesting study, which may raise the caution of targeting VCP alone for the therapeutics of tauopathies. I have a few comments on this manuscript.

1. The authors carried out excellent and detailed mechanistic studies mostly in cultured cell lines and a few in primary neurons, however, they did not show any evidence from animal models or human disease-related models. Thus, the reviewer is not sure how these findings in vitro can be applied in vivo and/or in humans.
2. The authors demonstrated the effect of the VCP disease mutations on disaggregation, if any, is only mild, suggesting that inhibition of aggregate clearance may not be the primary mechanism by which these mutations cause disease. This conclusion should be cautious since all the data are based on the studies using HEK293T cells. The authors may at least validate it using primary neurons. Also, the authors may discuss more about previous reports on VCP mutations and potential mechanisms.
3. Although the authors showed that the effects of VCP is macroautophagy independent, how about microautophagy or chaperone-mediated autophagy (CMA)? The authors did not show any data or discuss them. Also, previous studies have shown the macroautophagy could be one potential mechanism underlying the function of VCP. The authors should at least discuss this inconsistency between their data with others.

Reviewer #4 (Remarks to the Author):

Saha et al. describe an alternative mechanism of the tau fibril clearance by VCP chaperone which binds to ubiquitinated tau fibrils and recruits them for degradation by the proteasome. Using both the fluorescently labeled aggregation-prone region of tau and the full-length tau, the authors conducted experiments in HEK293T cells and in primary neurons which support their hypothesis and show that VCP is involved in tau fibril clearance. Overall, the manuscript is very well written and provides a large number of experiments to support the VCP role which lays a good ground for further work (by authors or others) to further confirm the VCP function using structural biology methods. This reviewer thinks that structural data describing the mechanism of VCP-Tau interaction at the molecular level would greatly improve the manuscript.

The methods section provides sufficient level of details.

This reviewer has the following recommendations to further improve the quality of the manuscript.

Major comments:

1. The authors discuss the role of VCP mutation in neurodegenerative diseases. In the light of the recent data from Shi et al. (Nature, 598), the tau fibril structure varies among different tauopathies. This may further confine VCP interaction and the mutation in VCP may not be the defining cause for the particular tauopathy. The discussion in this manuscript should reflect that.
2. The discussion about the potential role of VCP-Tau fragments forming the seeds for further fibrillization in other cells is not clear to this reviewer. Did the authors carry out experiments that such fragments can spread among the cells and seed further fibrillation?
3. The authors mention in the methods that they have carried out cryo-CLEM experiments. However, no cryo-CLEM data are shown in the manuscript. Actually, the correlation with fluorescence imaging would greatly improve the interpretation of Fig. 1d and 3c. In both cases, the authors denote the fibrillar objects in cryo-ET data as tau fibril. Although highly likely, this should be supported either by a control experiment or even better correlative imaging. In addition, Fig. 1d (right panel) gives a false notion that the filaments are restricted to a particular region of the depicted volume. There are obvious filaments of the same size in the top part of the image which are not shown in segmentation.

Minor comments:

1. "de novo", "in vitro" etc. shall be typeset in italic

Point-by-point response to reviewer comments

(all Page und line numbers in this response regard the document “Saha et al_Main-20220923_Plain Text_No edits.docx”)

Reviewer #1 (Remarks to the Author):

In this manuscript, Saha et al. carry out experiments to demonstrate that the AAA+ chaperone VCP (p97) binds ubiquitylated Tau fibrils and disaggregates them. They demonstrate that inhibition of VCP activity stabilizes large Tau aggregates; however, this also results in the reduction in the amount of Tau species competent of prion like aggregate seeding in recipient cells. Therefore, the authors conclude that disaggregation by VCP generates seeding-active Tau as byproduct. Proper aggregate clearance requires the functions of Hsp70 and of the UPS system.

General impression

This is a very detailed study that presents a lot of data using multiple systems and multiple approaches to support the claims. However, it is mainly based on phenotypic observations. The authors have not done any of the needed in vitro experiments to show that isolated Tau fibrils can be disassembled by purified VCP. Furthermore, the role of Hsp70 is no well investigated and seems to be a side note in the manuscript.

We thank the reviewer for these favorable comments.

This is the first study providing direct evidence that VCP in cooperation with the proteasome system mediates the disaggregation and disposal of Tau fibrils in cells. This activity has so far mostly been attributed to the Hsp70 chaperone system, although mainly on the basis of in vitro experiments.

In this manuscript we now show in cells that the system necessary for Tau disaggregation and degradation is very complex, and consists of at least three major machineries: VCP and its cofactors, the ubiquitin-proteasome system and the Hsp70 system. The Tau aggregates to be disaggregated must be modified by ubiquitylation by an E3 ligase(s) that remains to be

identified. Reconstitution of this process *in vitro* with purified components is highly challenging and is therefore outside the scope of this first study. One particular challenge, apart from having to express and purify VCP, cofactors and proteasome in functional form, is to produce the properly ubiquitylated Tau aggregates.

We agree with the reviewer that Hsp70 is mainly a side-note in this project, but we believe that the experiments presented are useful in providing a link to published literature. We have now included a sentence in the discussion (page 15, lines 332-334) to emphasize that due to the structural diversities of Tau fibrils, both the VCP pathway, described here, and a Hsp70 mediated reaction of disaggregation can exist in cells.

Specific comments

1. The authors should quantify the inclusions/cell for Fig. 2d.

The quantification of inclusions/cell is now shown in Fig. 2d. VCP and proteasome inhibition significantly stabilize aggregates in Tet-TauRD-Y* cells.

2. No filter-trap analysis done for CB-5083?

We added the filter-trap analysis for CB-5083 as Supplementary Fig. 5e. We observe similar effects of inhibiting VCP using CB-5083 as with NMS-873.

3. Filter-trap analysis of siVCP in Supplementary Fig. 4f are not very convincing, perhaps due to low efficiency of siRNA-mediated knockdown.

Yes, since VCP is essential for cell survival, it is indeed not trivial to deplete it completely from cells. To show the effect of VCP knockdown on the filter trap assay more clearly, we have added a quantification of the data in Supplementary Fig. 5g.

4. Line 171-172 vs line 203: Is the TauRD cell line tagged with myc or untagged? They are referred to differently in the two places indicated. If myc-tagged, naming conventions should be kept consistent, and should be renamed TauRD-M instead.

The TauRD cell line has a C-terminal myc tag as stated in line 173. This is now also indicated in line 213/214. We added new schematics to Fig.1a now including the non-fluorescent tagged FLTau and TauRD constructs.

5. Claims of EQ/EQ mutants effects on TauRD-Y aggregates from lines 261-264 is not backed by actual data.

We agree and have omitted this statement.

6. Is there a reason that the AE and RH mutants are not analyzed for whether they stabilize aggregates as the DG and EQ/EQ mutants in Fig. 5d? AE seems to colocalize nicely with aggregates in Fig. 5b.

We added the analysis of AE and RH mutants in Fig. 5d. We do not observe a significant stabilization of TauRD-Y aggregates upon expressing these mutants. This observation is consistent with the filter-trap analysis shown in Fig. 5c and current understanding that these mutations rather increase the ATPase activity and unfolding capacity of VCP.

Reviewer #2 (Remarks to the Author):

Thank you for writing a very comprehensive paper

A few general comments:
Lovely microscopy images!

We thank the reviewer for her/his positive comments.

Perhaps add into the title: "The AAA+ chaperone VCP disaggregates Tau fibrils and generates aggregate seeds " that this was a laboratory model

We have changed the title to "The AAA+ chaperone VCP disaggregates Tau fibrils and generates aggregate seeds in a cellular system" to emphasize that these experiments were done in a laboratory model.

Please write out all abbreviations in the abstract for general readers who are not familiar with the specific field: eg. valosin-containing protein (VCP)

We thank the reviewer for pointing this out. We have now written out VCP and Hsp70 in the abstract.

In the abstract you state:

These findings identify VCP as a core component of the machinery for the removal of neurodegenerative disease aggregates and suggest that its activity can be associated with enhanced aggregate spreading in tauopathies.

Please add to your discussion how you relate your findings in the lab to what happens during neuro-inflammation in AD. Perhaps a figure could be added to the explanation.

The reviewer has raised an interesting idea connecting our findings to neuroinflammation, which contributes critically to AD pathology. The general consensus about the contribution of protein aggregates to neuroinflammation in AD is that CNS-resident microglia recognize extracellular amyloid- β aggregates and Tau 'ghost tangles' via cell surface receptors and are activated by this process. Activated microglia secrete pro-inflammatory molecules and trigger reactions of the immune system leading to neuroinflammation. We surmise that disaggregated Tau seeds once released to the extracellular space, in addition to spreading aggregation to neighboring cells, may activate microglia and trigger an immune reaction,

thereby contributing to the characteristic neuroinflammation observed in AD. We have included this idea in the discussion (page 17, lines 365-368) of the revised manuscript.

VCP-mediated aggregate disassembly followed by proteasomal degradation provides an important alternative to autophagy as a mechanism for the elimination of terminally aggregated proteins.

Please add a diagram to explain autophagy and compare it to your VCP-mediated pathways and findings.

Autophagy is an important and complex cellular pathway that has been extensively studied and reviewed in the literature. Due to space limitations, we feel a review article would be the more appropriate format to elaborate on the interplay between VCP-mediated disaggregation, aggregate clearance by autophagy and other direct and indirect roles of VCP in the cell.

Please relate your findings directly to clinical application in AD. How do you foresee this research may impact clinical interventions?

We appreciate that the reviewer considers the possibility that our findings may impact clinical interventions in AD in the future. We currently state that (page 17, lines 371-376) 'Based on our results, both activation and inhibition of this pathway (i.e. disaggregation by VCP) may have beneficial effects dependent on the specific disease context. Non-human AAA+ ATPases with augmented disaggregase activity are currently being developed with the aim to reverse pathogenic protein aggregation. Boosting cellular aggregate clearance, perhaps in combination with proteasome activation, may offer a potential therapeutic strategy as long as the production of seeding competent species can be controlled.' Since VCP plays a role in different cellular pathways, we feel that larger claims may raise unfounded public expectations about the time necessary to transfer these results to a clinical setting.

Please elaborate or discuss in the conclusion if this work could be translated to other neuro-inflammatory diseases too? Parkinson's Disease?

Given that VCP colocalizes with Lewy bodies in patient brains from Parkinson's disease and dementia with Lewy bodies (Hirabayashi et al. Cell Death and Differentiation 2001, PMID: 11598795), it is possible that VCP is involved in modulating these aggregates as well. However, this activity has not yet been demonstrated. Also, it is unclear whether VCP would directly disaggregate Lewy body-resident α -synuclein aggregates or mediate its action via its role in autophagy. This is critical because the underlying molecular mechanism of VCP function would dictate whether or not seeding-competent species with possible downstream effects are generated. Future experiments with other disease-related aggregates and in relevant model systems will be necessary to test the generality of VCP-mediated disaggregation and the contribution to neuroinflammation. We believe that at this point it would be premature to speculate on the outcome of the interactions between alpha-synuclein aggregates and VCP.

Reviewer #3 (Remarks to the Author):

Saha et al. reported VCP disaggregates Tau fibrils, however, it generates seeding-active Tau as byproduct at the same time. These results suggest that VCP may have both neuroprotective and neurotoxic effects, although VCP mutations have been associated with aggregate deposition disorders such as vacuolar tauopathy and IBMPFD. Boosting cellular aggregate clearance, perhaps in combination with proteasome activation, may offer a potential therapeutic strategy as long as the production of seeding competent species can be controlled. This is an interesting study, which may raise the caution of targeting VCP alone for the therapeutics of tauopathies. I have a few comments on this manuscript.

1. The authors carried out excellent and detailed mechanistic studies mostly in cultured cell lines and a few in primary neurons, however, they did not show any evidence from animal models or human disease-related models. Thus, the reviewer is not sure how these findings *in vitro* can be applied *in vivo* and/or in humans.

We thank the reviewer for his/her positive comment. In this first study demonstrating VCP-mediated disaggregation of Tau fibrils, we have limited our experiments mainly to cultured mammalian cells and mouse primary neurons. In the revised draft, we have added new data from the tauopathy mouse model rTg4510 expressing human Tau with the P301L point mutation, where we tested whether VCP colocalizes with Tau aggregates in brain sections. Indeed, immunohistochemistry experiments show that VCP colocalizes with hyperphosphorylated Tau aggregates in the mouse brain. This result is now shown in Supplementary Fig. 6g. These observations further consolidate our findings in cultured cells. We have now emphasized the limitations of our model system in the discussion (page 16 line 340-341; page 17, line 353-354), and also altered the title of the manuscript to make this point clearer. Future studies will focus on a deeper investigation of VCP function in disaggregation in the *in vivo* model, but are outside the scope of the present manuscript.

The important role of VCP in modulating Tau aggregation in human brains is supported by a newly described neurodegenerative disease called vacuolar tauopathy, where patients carrying a hypomorphic VCP mutation, D395G, develop Tau aggregates in the brain (Darwich et al., Science 2020, PMID: 33004675). Furthermore, VCP colocalizes with Tau aggregates in Alzheimer's disease (Darwich et al., Science 2020, PMID: 33004675), inclusions of expanded polyglutamine protein huntingtin in Huntington's disease and Lewy bodies comprised of α -synuclein in Lewy Body Dementia (Hirabayashi et al. Cell Death and Differentiation 2001, PMID: 11598795). Therefore, it is plausible that VCP has a common function in protein disaggregation relevant to a wide range of human diseases - a topic for further investigation.

2. The authors demonstrated the effect of the VCP disease mutations on disaggregation, if any, is only mild, suggesting that inhibition of aggregate clearance may not be the primary mechanism by which these mutations cause disease. This conclusion should be cautious since all the data are based on the studies using HEK293T cells. The authors may at least validate it using primary neurons. Also, the authors may discuss more about previous reports on VCP mutations and potential mechanisms.

We agree with the reviewer that neurons may be more sensitive to VCP mutation than HEK293T cells. Accordingly, we have revised our conclusion (page 13, line 275-276) and extended the discussion on VCP mutations and potential mechanisms (page 16, lines 347-350). However, validating the mutants in neurons will require a substantial amount of testing and optimization. Such experiments, while not central to our main conclusions, would unduly delay publication.

3. Although the authors showed that the effects of VCP is macroautophagy independent, how about microautophagy or chaperone-mediated autophagy (CMA)? The authors did not show any data or discuss them. Also, previous studies have shown the macroautophagy could be one potential mechanism underlying the function of VCP. The authors should at least discuss this inconsistency between their data with others.

Emerging evidence indeed suggests that VCP also plays a role in autophagy. However, as shown in Supplementary Fig. 4a, we did not observe TauRD-Y stabilization upon inhibiting lysosomal degradation with BafilomycinA1, where all forms of autophagic degradation should be blocked, including microautophagy and CMA. We have clarified this in the text (page 7, lines 133-136). The independence of autophagy, whether or not a peculiarity of our model system, provides the advantage to unequivocally identify the function of VCP in disaggregation.

Reviewer #4 (Remarks to the Author):

Saha et al. describe an alternative mechanism of the tau fibril clearance by VCP chaperone which binds to ubiquitinated tau fibrils and recruits them for degradation by the proteasome. Using both the fluorescently labeled aggregation-prone region of tau and the full-length tau, the authors conducted experiments in HEK293T cells and in primary neurons which support their hypothesis and show that VCP is involved in tau fibril clearance. Overall, the manuscript is very well written and provides a large number of experiments to support the VCP role which lays a good ground for further work (by authors or others) to further confirm the VCP function using structural biology methods. This reviewer thinks that structural data describing the mechanism of VCP-Tau interaction at the molecular level would greatly improve the manuscript.

The methods section provides sufficient level of details.

We thank the reviewer for her/his positive comments, and agree with the assessment that understanding, in structural terms, how VCP interacts with Tau (and many of its other substrates) will be important. However, such experiments are clearly outside the scope of this first article on the mechanism of VCP-mediated fibril disaggregation and will require a separate study.

This reviewer has the following recommendations to further improve the quality of the manuscript.

Major comments:

1. The authors discuss the role of VCP mutation in neurodegenerative diseases. In the light of the recent data from Shi et al. (Nature, 598), the tau fibril structure varies among different tauopathies. This may further confine VCP interaction and the mutation in VCP may not be the defining cause for the particular tauopathy. The discussion in this manuscript should reflect that.

We agree and thank the reviewer for this comment. We have altered our discussion to better reflect this, and have added a statement that different forms of Tau aggregate may be more or less suitable for VCP-mediated disaggregation (page 15, line 332-334).

2. The discussion about the potential role of VCP-Tau fragments forming the seeds for further fibrillization in other cells is not clear to this reviewer. Did the authors carry out experiments that such fragments can spread among the cells and seed further fibrillation?

The experiments shown in Figures 6 and S12 have been mainly designed to detect oligomeric, seeding competent Tau species that when added to cells are imported and induce new Tau aggregates. However, the reviewer is correct in that we have not demonstrated that such oligomers or fragments can be exported from cells for uptake by neighboring cells. Further experiments will be required to investigate the mechanism of seed export and uptake. Such studies are ongoing in several laboratories. We discuss this now in more detail and point out the limitations of the present study (page 17, lines 361-365).

3. The authors mention in the methods that they have carried out cryo-CLEM experiments. However, no cryo-CLEM data are shown in the manuscript. Actually, the correlation with fluorescence imaging would greatly improve the interpretation of Fig. 1d and 3c. In both cases, the authors denote the fibrillar objects in cryo-ET data as tau fibril. Although highly likely, this should be supported either by a control experiment or even better correlative imaging. In addition, Fig. 1d (right panel) gives a false notion that the filaments are restricted to a particular region of the depicted volume. There are obvious filaments of the same size in the top part of the image which are not shown in segmentation.

We thank the reviewer for pointing this out. To address this comment, we introduced a new Supplementary Fig. 2 showing a schematic of the different steps of the cryo-ET procedure that was carried out for imaging the

inclusions in HEK293 cells and neurons (Supplementary Fig. 2a), and specific images from the cryo-CLEM workflow of the inclusions in neurons (Supplementary Fig. 2b-g). Additionally, we are providing a similar figure of the cryo-CLEM workflow in HEK293 cells here for the reviewer's assessment (Fig. I). Correlative imaging, in addition to the presence of YFP densities on the fibrils as shown in Supplementary Fig. S6f (as shown in Bäuerlein et al., Cell 2017, PMID: 28890085; Guo et al. Cell 2018, PMID: 29398115; Trinkaus et al. Nat Comm 2021, PMID: 33854052), confirms that the fibrils are indeed formed of TauRD-Y protein.

Regarding the additional fibrils in Fig. 1d, we think that the structures that the reviewer is referring to might be membranes, as shown here in the overlay of the tomogram with the segmentation (Fig. If). Moreover, the fibrils traced in the segmentation are representative and meant only for visualization. We have not used this data for any quantitative analysis of fibril abundance or specific localization, and cannot exclude the possibility that there may be some fibrils in other areas of the cell. The raw tomograms corresponding to Fig. 1d and 3c are available at EMDB (EMD-13739 and EMD-13740, respectively) for the reference of the readers and reviewers.

Fig 1: Cryo-correlative-light-electron microscopy (cryo-CLEM) workflow of HEK293 cells.

a TauRD-Y* cells containing TauRD inclusions were cultured on EM grids for 24 h and vitrified by plunge freezing. Thereafter, grids were imaged by cryo-LM (cryo-light microscopy) and cryo-SEM (cryo-scanning electron microscopy), and lamellae were generated by cryo-focused ion beam milling. An overlay of cryo-SEM and cryo-LM images is shown. A cell of interest is marked by box L6. L2 shows an additional lamella. Scale bar, 30 μ m. **b** Magnified cryo-SEM image of the ~200 nm thick L6 lamella from (a). Scale bar, 15 μ m. **c** Cryo-SEM and cryo-LM overlay at the location of the L6 lamella. Scale bar, 30 μ m. **d** Cryo-TEM (cryo-transmission electron microscopy) overview of the lamella shown in (b, c). Tomograms were acquired in regions indicated by boxes. The tomogram shown in (e) was acquired in the area represented by the white box. IC: Ice crystal. Scale bar, 3 μ m. **e** 1.7 nm thick tomographic slice of a TauRD inclusion from TauRD-Y* cells (shown in Fig. 1d). Red, blue and green arrowheads indicate representative TauRD-Y fibrils, microtubule and actin, respectively. Scale bar, 300 nm. **f** An overlay of the 3D rendering with the tomogram in (e) showing TauRD-Y fibrils (red), Golgi (purple), mitochondria (yellow) and ER (green). Scale bar, 300 nm.

Minor comments:

1. "de novo", "in vitro" etc. shall be typeset in italic

We agree this is normally the case, however, we followed the non-italicized style of Nature Communications to write 'de novo', 'in vitro', etc.

REVIEWER COMMENTS

Reviewer #1 (Remarks to the Author):

The authors have addressed my comments. I have no further comments.

Reviewer #3 (Remarks to the Author):

The authors provided new data and more discussions in their revision. The current version of the manuscript has been improved compared to the original submission. But I have a question about their new data in Supplementary Fig. 6g: Immunohistochemical staining of brain section of a 4-month-old Tau transgenic rTg4510 mouse with AT8 (green) and VCP (red) antibodies, and Nissl substance (cyan).

First, the staining of VCP in AT8-positive neuron shows the pattern very similar to AT8, i.e. mainly puncta staining in the cytosol. However, the staining of VCP in AT8-negative neuron shows the diffused staining like the background staining. The authors need to show the VCP staining of non-transgenic control mice, which will let us know the normal staining pattern of VCP, and how it changes in rTg4510 mice. Also, did the authors validate the specificity of this VCP antibody?

Second, if VCP is essential for pTau disaggregation, shouldn't the authors see reduced VCP expression or activity in AT8-positive neurons? However, the authors' data seems to be opposite. Maybe the puncta of VCP does not indicate the higher level of VCP, but actually suggesting the dysfunction of VCP or reduced activity of VCP?

Reviewer #4 (Remarks to the Author):

The authors have satisfactorily addressed my original comments. I fully support the acceptance of the manuscript for publication.

REVIEWER COMMENTS

Reviewer #3 (Remarks to the Author):

The authors provided new data and more discussions in their revision. The current version of the manuscript has been improved compared to the original submission. But I have a question about their new data in Supplementary Fig. 6g: Immunohistochemical staining of brain section of a 4-month-old Tau transgenic rTg4510 mouse with AT8 (green) and VCP (red) antibodies, and Nissl substance (cyan).

First, the staining of VCP in AT8-positive neuron shows the pattern very similar to AT8, i.e. mainly puncta staining in the cytosol. However, the staining of VCP in AT8-negative neuron shows the diffused staining like the background staining. The authors need to show the VCP staining of non-transgenic control mice, which will let us know the normal staining pattern of VCP, and how it changes in rTg4510 mice. Also, did the authors validate the specificity of this VCP antibody?

Second, if VCP is essential for pTau disaggregation, shouldn't the authors see reduced VCP expression or activity in AT8-positive neurons? However, the authors' data seems to be opposite. Maybe the puncta of VCP does not indicate the higher level of VCP, but actually suggesting the dysfunction of VCP or reduced activity of VCP?

To address the reviewer's first comment, we have added panels showing VCP immunostaining in the brain section of littermate control mice (Supplementary Fig. 6g). Both in control and transgenic mice, diffuse VCP signal is observed in neurons not containing pTau aggregates. Therefore, the punctate appearance of VCP in pTau positive neurons of rTg4510 mice is due to interaction with Tau aggregates.

The antibody that we used for these stainings NB100-1558 (Novus Biologicals) has previously been used for mouse brain immunofluorescence (Clemen et al. *Brain*, 2010; PMID: 20833645). We further validated the specificity of this antibody by immunoblotting (IB) and immunofluorescence (IF) analysis in VCP knockdown samples from mouse and human cell lines. Specifically, we used immortalized mouse neuronal Neuro2a cells, mouse embryonic fibroblast (MEF) and human HEK293T cells where VCP was down-regulated using siRNA. IB analysis showed that the intensity of the recognized band decreased when cells were treated with the VCP siRNA (Fig. 1a), confirming that the antibody recognized VCP in all three cell lines. The knockdown is relatively weak in mouse cell lines due to the inherently low transfection efficiency of these cell lines. Furthermore, IF analysis of VCP showed cells with reduced fluorescence intensity only under VCP knockdown condition (Fig. 1b). Additionally, immunoblot analysis of mouse brain lysates using this antibody showed a single band consistent with the molecular weight of VCP (Fig. 11b). These experiments validate the specificity of this antibody in recognizing mouse and human VCP for IB and IF applications.

Figure I: Validation of VCP antibody NB100-1558 (Novus Biologicals).

a Analysis of VCP levels in Neuro2a, Mouse Embryonic Fibroblast (MEF) and HEK293T cells treated with control or VCP siRNA for 96 h. VCP was blotted using the antibody NB100-1558 (Novus Biologicals). Tubulin served as loading control. Percentage knockdown relative to control is indicated. **b** Immunofluorescence staining of VCP (green) in cells treated as in (a). White arrowheads indicate cells with reduced VCP levels. Scale bars, 10 μ m.

In the second point, the reviewer speculates that aggregate-containing neurons should have reduced VCP expression and that our data seems to be the opposite. While a (e.g. age related) reduction of VCP activity might contribute to a reduced Tau aggregate clearance in a disease context, we do not provide (and are not aware of) any evidence that Tau aggregation correlates with decreased VCP activity in tauopathy patients without VCP mutations. Indeed, VCP levels are not reduced in the human brain in AD (Bai et al. *Mol. Neurodegeneration*, 2021, PMID: 34384464), a fact that we now also mention in the manuscript. Note that in the tauopathy mouse model used the mutation is in Tau and there is no reason to assume that VCP levels are altered compared to control animals. In order to address the reviewer's question, we now quantified VCP fluorescence intensity in pTau+ and pTau- neurons in rTg4510 mouse brain. A mild but non-significant reduction

of VCP levels was observed in pTau+ neurons (Fig. IIa). We further compared VCP levels in lysates from the cortex of 4 - 4.5 months old control and rTg4510 mice. In line with the IF result, IB analysis showed no significant difference in the VCP levels between control and rTg4510 mice (Fig. IIb). Notably, a knock-in mouse model expressing hypomorphic VCP mutant D395G shows no Tau pathology (Darwich et al., Science, 2020, PMID: 33004675), indicating that reduced VCP activity is not sufficient to trigger Tau aggregation. Our results with NMS-873 treated cells are consistent with this idea (Supplementary Fig. 5i and 6c). Aggregates accumulate presumably due to overwhelmed disaggregation capacity of the cell. Additionally, a fully functional aggregate-clearance pathway requires the downstream Hsp70 chaperone system, the proteasome and under some conditions, autophagy. Perturbation of these pathways is often observed neurodegeneration models and likely to contribute to cellular aggregate load.

Assessing whether the aggregate-localized VCP is dysfunctional or actively disaggregates Tau in neurons *in vivo* is highly challenging and outside the scope of this manuscript. Notwithstanding, it is an exciting question and we are actively investigating this in living cells in an independent project using advanced microscopy techniques and cell culture models. In the present manuscript, we showed that disaggregation is blocked when VCP is not localized to the inclusions (Fig.4c,d), therefore we think that the presence of VCP on aggregates is meaningful.

Figure II: Analysis of VCP levels in rTg4510 mice.

a Quantification of VCP fluorescence intensity in AT-8 negative (pTau-) and AT-8 positive (pTau+) neurons in rTg4510 brain sections (related to Supplementary Fig. 6g) co-stained with a validated VCP antibody NB100-1558 (Novus Biologicals). Mean \pm s.d.; $n=3$ mice, 50 neurons/mouse; n.s. non-significant ($p= 0.0616$) from unpaired t-test. **b** Left, immunoblot analysis of lysates from the cortex of control and rTg4510 mice. VCP was blotted using the antibody NB100-1558 (Novus Biologicals). GAPDH served as loading control. Right, quantification of VCP band intensity. Mean \pm s.d.; $n=4$ mice. n.s. non-significant ($p= 0.2920$) from unpaired t-test.

REVIEWERS' COMMENTS

Reviewer #3 (Remarks to the Author):

The authors have provided new data and discussions to address the reviewer's comments. I would like to recommend the acceptance of the manuscript for publication in Nature Communications.